# One-year in: COVID-19 research at the international level in CORD-19 data

**Caroline S. Wagner**[1]*, **Xiaojing Cai**[2], **Yi Zhang**[3], **Caroline V. Fry**[4]

**1** John Glenn College of Public Affairs, The Ohio State University, Columbus, Ohio, United States of America, **2** School of Public Affairs, Zhejiang University, Hangzhou, Zhejiang, China, **3** Australian Artificial Intelligence Institute, University of Technology Sydney, Ultimo, Australia, **4** Shidler College of Business, University of Hawaiʻi at Mānoa, Honolulu, Hawaiʻi, United States of America

* wagner.911@osu.edu

**Data Availability Statement:** https://doi.org/10.6084/m9.figshare.16620274.v1.

**Funding:** The authors received no specific funding for this work.

## Abstract

The appearance of a novel coronavirus in late 2019 radically changed the community of researchers working on coronaviruses since the 2002 SARS epidemic. In 2020, coronavirus-related publications grew by 20 times over the previous two years, with 130,000 more researchers publishing on related topics. The United States, the United Kingdom and China led dozens of nations working on coronavirus prior to the pandemic, but leadership consolidated among these three nations in 2020, which collectively accounted for 50% of all papers, garnering well more than 60% of citations. China took an early lead on COVID-19 research, but dropped rapidly in production and international participation through the year. Europe showed an opposite pattern, beginning slowly in publications but growing in contributions during the year. The share of internationally collaborative publications dropped from pre-pandemic rates; single-authored publications grew. For all nations, including China, the number of publications about COVID track closely with the outbreak of COVID-19 cases. Lower-income nations participate very little in COVID-19 research in 2020. Topic maps of internationally collaborative work show the rise of patient care and public health clusters—two topics that were largely absent from coronavirus research in the two years prior to 2020. Findings are consistent with global science as a self-organizing system operating on a reputation-based dynamic.

## Introduction

The COVID-19 pandemic upended many normal practices around the conduct of research and development (R&D); the extent of disruption is revealed across measures of scientific research output [1–3]. This paper revisits the extent to which patterns of international collaboration in coronavirus research during the COVID-19 pandemic depart from 'normal' times. We present publication patterns using one full year of publications data from the CORD-19 database, and observations on non-COVID peer-reviewed publications using the Web of Science, to examine national and international publication rates and network patterns. We examine topics of research on COVID-19, and reflect on lessons learned about international

**Competing interests:** The authors have declared that no competing interests exist.

collaboration from the disruption. The analysis may be useful to research administrators, international affairs professionals and science studies scholars.

We study the international collaborative linkages as a network. In the absence of a global governing body, international collaborations operate by network dynamics. Scientific connections at the global level reflect collective decisions of hundreds of individuals who seek to connect to each other. Connections are not random; they are influenced by five factors: two personal and three contextual. Personal choices tend towards those previously known or known by reputation or introduction. Contextual choices are 1) resources available, 2) geopolitical factors, and 3) time and attention. Network dynamics emerge from interplay of these factors, although there is little research on how a disaster, such as a pandemic, will affect productivity, collaboration, and topic focus. Moreover, it is difficult to determine expectations of network dynamics in a pandemic because global exogenous disruptions are rare and studies about science in a disaster are sparse. This paper seeks to fill some of these gaps.

This paper is organized to describe the literature supporting our inquiries, and to present hypotheses derived from the literature. We then describe coronavirus research prior to the pandemic, and early policy responses to the crisis. A section on data and methodology presents approaches designed to answer the questions emerging from the hypotheses. A results section describes outcomes of the analyses, followed by limitations of the data and approaches presented here. A discussion section details responses to the hypotheses as well as observations about the research project and avenues for further research. An S1 Appendix provides additional technical details.

## Literature review and hypotheses

In the decades preceding the pandemic, R&D spending and output grew rapidly. OECD data shows that, among member nations, R&D spending was 25% higher in 2017 than a decade earlier. The US National Science Foundation (NSF) reports that from 2008–2018 the annual number of citable publications (articles, notes and letters, hereafter, "publications") worldwide grew by 3.83% per year from 1.8 million to 2.6 million [4]. Increases in spending, trained practitioners, and publications contributed to an overall growth of the research enterprise in natural sciences and engineering, social sciences, and arts and humanities. Within the research enterprise, among scientifically advanced nations, international collaborative publications grew at a faster rate than national publications, accounting for as much as one-quarter of all publications in 2018, with variations observed across fields, according to the National Science Foundation [4]. Those fields that rely on large-scale equipment are more highly globalized, but increases in international linkages is observed in most fields, tied, not to funding or equipment, but to the interests of researchers to work together. The size of these teams has grown larger over time [5].

International collaborative patterns have been dominated by scientifically advanced nations, although, over time, many low-income, emerging and developing nations have become more active, and have partnered with more advanced nations [6]. Some tendency to collaborate among nations with former colonial ties is observed [7], but this is likely due to incentivized funding provided by the former colonial power. Political differences do not appear to hinder collaboration, evidenced most notably by the rise of China to be the number one collaborating nation with the United States. Abramo et al. [8] added to literature on tendency of neighbors to work together, but Choi [6] shows this tendency to be decreasing over time.

Prior research into collaboration around viral disease events found that, during the 2014 West African Ebola epidemic, collaboration grew between scientists from scientifically

advanced nations and the most affected nations [9], suggesting that connections were made based upon disease location. Ebola outbreaks brought in researchers from scientifically advanced nations to work with local researchers on specific events. Collaborative ties did not persist past the disease event.

A global community of coronavirus researchers predated the advent of the 2019 novel coronavirus; this community formed after the 2002 SARS coronavirus epidemic [1]. As the new threat emerged in 2019, governments provided emergency R&D funding to encourage targeted research on the novel coronavirus. Most of these funds were committed by governments in scientifically advanced countries and were allocated to national institutions, although the European Union (EU) and the US National Institutes of Health (NIH) fund both national and foreign applicants. National actors receiving funds may then choose, in some instances, to connect to foreign collaborators, creating an international connection. The resulting connections can be studied through coauthorship attributions on paper and interpreted as a self-organizing network of connectivity [10]. In Fry et al. [1], we showed that, during the early months of the COVID-19 pandemic, international collaborations in coronavirus research emerged among just a few nations, and, on average, publications had fewer coauthors per paper than pre-pandemic levels. Most nations did not publish on the novel coronavirus in early pandemic research.

We expect that, as funds became available to researchers, and as more knowledge was generated through the first year of the pandemic, cross-national collaborative ties will grow. That said, because of travel limitations and a need for urgent results, we expect the rate of international collaboration and network ties to remain lower than pre-pandemic levels. This expectation is also informed by the research of Rotolo and Frickel [11] who found that there were fewer ties and smaller teams among researchers just after a hurricane disaster. Further, based upon findings in the wake of the Fukushima disaster [12] and a survey by Myers et al. [2] we expect that attention to pandemic-related R&D (including basic science, patient care, and public health) has lessened the output of other scientific research as well as reduced the rate of international collaborations in other fields. In addition to changed collaborative patterns, we expect to see changes in topics throughout the first year of the pandemic with topics becoming more focused as knowledge about events grows, which we explore in a separate article. In Zhang et al. [3], we showed that, at the beginning of the pandemic, the disrupted knowledge system exhibited very little topic focus. As the pandemic progresses, we expect to see greater topic focus. We further expect to see international collaboration focus on basic science and less on patient care and public health which may have a local, regional, or national characteristics. We expect continued consolidation among leading nations and elite institutions through the pandemic year due to pressures for rapid results and the lack of mobility to begin new collaborations. Further, we expect that geographic distance will mean less during the pandemic because remote collaborators will rely on communications technologies rather than face-to-face consultations.

## Science during the COVID-19 pandemic

Coronavirus research predated the COVID-19 crisis, but it was a community of 22,000 researchers working on SARs, MERs, and the porcine diarrhea epidemic [1, 3]. Coronavirus research output doubled in number over the decade between 2008–2018, in keeping with numbers in the biological sciences. As the new threat of a novel coronavirus emerged in 2019, governments provided emergency R&D funding to encourage targeted research, which attracted many new researchers from a wide range of fields. More than 156,000 researchers published on COVID-19 in 2020, growing the original community that had worked on coronaviruses by over 130,000.

The United States Government committed the largest amount of funds to the novel coronavirus, through the CARES Act and other legislation, allocating at least $5 billion to basic research, applied research, and development of vaccines, diagnostics, mapping of disease occurrence, analytics, public health, and medicine. The bulk of funds were appropriated by the US Congress to the U.S. National Institutes of Health (NIH), and through them to BARDA, the Biomedical Advanced Research and Development Authority. Other agencies also received additional R&D funds over and above their annual appropriations, including the National Science Foundation ($74 million) and the Department of Energy ($99.5 million). The U.S. government also provided funds to private companies to aid in vaccine development and procurement. For example, Moderna, a pharmaceutical company headquartered in Massachusetts, received $1 billion of R&D funds and in $1.5 billion in advanced purchase agreements.

Germany provided $891 million in R&D funds into coronavirus as well as to vaccine development. The European Commission provided €469 million in R&D funds, along with permission to recipients to reallocate funds originally slotted for other topics. The UK government reports spending £554 million on 3,600 initiatives related to COVID-19. In China, the Ministry of Science and Technology invested $100 million for emergency projects and unknown millions of funds for vaccine development, and the National Natural Science Foundation of China also reallocated approximately $15 million for projects related to COVID-19.

Many research organizations and researchers from various disciplines shifted to focus on aspects of the pandemic and received grant funds to do so. Just as with any other R&D funding, the expectation is that funded research will result in published works, enhanced equipment, and medicines and vaccines. Very early in the pandemic, preprints [13] (non-peer-reviewed articles) and peer reviewed articles began flooding into publishing venues. The number of scholarly publications related to the crisis grew spectacularly in the early months of the pandemic [1].

The rush to publish is expected: Zhang et al. [14] note that historical patterns show that researchers have, in previous cases, responded quickly to public health emergencies with publications, which is the same pattern we see with COVID-19 research. In updating our earlier work [15], we found that the number of coronavirus publications in CORD-19 grew considerably in the early days of the novel coronavirus, rising at a spectacular rate from a total of 4,875 articles produced on the topic (preprint and peer reviewed) between January and mid-April to an overall sum of 44,013 by mid-July, and accumulated to 87,515 by the start of October 2020. (In comparison, nanoscale science was a rapidly growing field in the 1990s, but it took more than 19 years to go from 4,000 to 90,000 articles [16]).

The dissemination of publications changed during the pandemic. COVID-19 peer-reviewed and edited publications became available to other researchers through new (CORD-19) and pre-existing (National Library of Medicine) web platforms. COVID-19 publications were much more likely than other works to be published as open access in 2020 [17]. In 2020, open-access, peer-reviewed publications related to COVID-19 accounted for 76.6% of all publications compared to 51% of all non-COVID publications. Highly cited papers—those in the top 1% most highly cited, with over 500 citations—were more likely than other work to be published in subscription-based journals such as *The Lancet*, *Science*, *New England Journal of Medicine* or *Nature* but these works were placed into open Web portals for rapid access. The National Library of Medicine served as a repository for most new publications related to COVID-19. The publishing house Elsevier—which publishes many subscription-based journals—created a "Public Health Emergency Collection" to make COVID-19 articles rapidly available regardless of the access status of the original work (subscription or open access). Similarly, CORD-19 (the database which provided data for this article) through Semantic Scholar,

made relevant research (including historical work) rapidly and readily available and allowed researchers to deposit work they viewed as relevant.

Researchers from China and the USA increased the rate of collaborative publications on coronavirus in the earliest days of the pandemic Fry et al. [1]. Liu et al. [18] showed a surge of what they call 'parachuting collaborations'–new connections not seen prior to the pandemic–which dramatically increased during the pandemic. Together with the findings in Fry et al. [1], these findings suggests that search and team formation changed to adapt to the needs of COVID-19 research, a finding also reported by Lee & Haupt [19]. Liu et al. [18] found that COVID-19 research papers were less likely to involve international collaboration than non-COVID-19 papers during the same time period, a finding reported by Aviv-Reuven & Rosen-feld [20] as well, a finding we can confirm.

Several research articles note the absence of emerging and developing nations in early COVID-19 research. Fry et al. [1] and Lee & Haupt [19] showed that very few developing nations were involved in the earliest day of the crisis. Zhang et al. [3] confirmed Fry et al. in finding that the USA, China, and the UK were the three countries with the largest number of articles by mid-year. Several articles report that fewer coauthors appear on article bylines [1, 20]. This is likely due to the need for rapidity in responding to the crisis: fewer coauthors reduces the time needed to communicate, synthesize and submit results.

## Data and methodology

Data for this study were extracted in March 2021 from the Covid-19 Open Research Dataset, "CORD-19," an open resource of scientific papers on COVID-19 and related historical corona-virus research. CORD-19 is designed to facilitate the development of text mining and information retrieval systems for COVID-19 research over its rich collection of metadata and structured full-text papers. It is accessible through the National Library of Medicine, National Institutes of Health, USA. In addition, we accessed the whole of Scopus 2020 data to examine non-COVID publications over the year. To search for evidence of government funding for COVID-19 research, we searched Web of Science, which has a field for funding acknowledge-ments. (Non-COVID publications were any peer-reviewed, published work that did not include one of the keywords for the COVID search below).

To maintain consistency across our studies, we applied the same search terms as used in Fry et al. [1], Cai et al. [15], and Zhang et al. [3] and limited the search to the dates January 2020 to December 2020 and citation data to March 2021. The following search terms were applied to titles and abstracts to obtain an initial dataset of coronavirus publications:

- COVID-19
- 2019-nCoV
- coronavirus
- corona virus
- SARS-CoV
- MERS-CoV
- Severe Acute Respiratory Syndrome
- Middle East Respiratory Syndrome

The initial dataset was cleaned to remove the following artifacts: conference papers, pre-prints, collections of abstracts, symposia results, articles pre-dating 2020, and meeting notes.

**Table 1. Number of coronavirus publications in 2020.**

|                                      | 2020 total | 2020 Q1 | 2020 Q2 | 2020 Q3 | 2020 Q4 |
|--------------------------------------|------------|---------|---------|---------|---------|
| Articles                             | 106,993    | 6,650   | 32,384  | 34,696  | 33,263  |
| Articles with address                | 92,008     | 4,758   | 27,333  | 30,716  | 29,201  |
| Sole-author articles                 | 8,158      | 556     | 2,880   | 2,576   | 2,146   |
| National collaborative articles      | 63,647     | 3,288   | 18,762  | 21,173  | 20,424  |
| Internationally collaborated articles| 20,203     | 914     | 5,691   | 6,967   | 6,631   |
| Rate of International collaboration   | 22.0%      | 19.2%   | 20.8%   | 22.7%   | 22.7%   |

Preprints were excluded in this report to avoid double-counting in cases where a work is sub-sequently peer-reviewed and published. The author names, institutional affiliation, and addresses were extracted for analysis. For articles derived from the PubMed Central website, the citation count was extracted up to March 2021. The resulting dataset provided us with 106,993 publications for the calendar year 2020. The final dataset was further divided into four quarters, shown in Table 1, according to "Published Date", i.e., the electronic publication dates (if any) or else print publication date: January to March (2020 Q1), April to June (2020 Q2), July to September (2020 Q3), and October to December (2020 Q4). Full counting is used to count the number of publications of a specific country or institution. Among all the publications with at least one author and address, 8,158 (8.9%) are single-author articles, with the rest involving coauthors at the national (78%) or international levels, with 20,203 (22.0%).

We analyzed the number of authors and coauthors per paper for descriptive statistics of people publishing on the novel coronavirus; we analyzed keyword usage and topics drawn from keywords and abstracts, and we analyzed geographic location of authors to study cooperative patterns at the international level. We collected additional data to answer questions about activities not available in CORD-19 in funding and on non-COVID research publications. We compared the CORD-19 data to a defined dataset of coronavirus research derived from scientific articles on coronavirus-related research on historical data we had earlier extracted from PubMed, Elsevier's Scopus, and Web of Science (details of the construction of this data can be found in Fry et al. [1]). The datasets are available at https://figshare.com/articles/dataset/One_Year_of_COVID-19_Int_l_Collaboration/16620274. For the CORD-19 articles that are also indexed in Clarivate's Web of Science (WoS), we retrieved funding information to get a rough view about which funding agency is contributing to the coronavirus research in the pandemic period; 35% of CORD-19 articles acknowledged funding. Dimensions database was used to analyze the open access categories.

To test for change in the number of participants in research groups between pre-COVID-19 and COVID-19 periods, we use double-tailed T-tests to compare the average "team" structure between periods. (We employ the word "team" for convenience to describe coauthor groups even though we do not know the mechanism of cooperation among the group.) Statistical significance is assessed at 0.05 level. Team structure is measured as average number of authors per publication, average number of nations per publication, and the percentage share of internationally collaborated articles. We also use regression models to test the relationship between team structure and citation impact. Since the dependent variable, i.e., citations, is a non-negative integer, we apply count-data regression models (i.e., negative binomial regression) that can account for the nature of the data.

In order to compare network structures of nations between pre-COVID-19 and during COVID-19, we construct global collaboration networks based on coauthorships in publications for the subset of data at the international level. Collaboration links among nations are

established based on author affiliations using a full counting method. For example, for an article with authors from the USA, Italy, and China, there is a single collaboration link between the USA and Italy, the USA and China, and Italy and China. We aggregate the number of ties over the publications in each period, and then use the R package "igraph" to compute the network metrics of global network and each node and visualize the collaboration networks using VOSviewer developed by van Eck and Waltman [21]. Measures taken are density—to examine the growth of interconnections across nations—and betweenness centrality—to assess the power relationships across countries in providing and sharing knowledge. Betweenness centrality measures the importance of a node in determining the flow of a network [22], and thus, in this study a high value of betweenness centrality indicates a crucial role in leading international collaborations. We particularly exploit a weighted betweenness centrality [23] to highlight the weight of edges (i.e., the frequency of collaboration) when calculating the shortest distance between two nodes. The equation for calculating betweenness centrality $bc(v_i)$ for node $v_i$ is given as follows:

$$bc(v_i) = \frac{2 \sum \frac{d(v_i)_{v_m, v_n}}{d_{v_m, v_n}}}{(V-1)(V-2)}, \ v_i \neq v_m \neq v_n$$

where $V$ denotes the total number of nodes in a network, $v_m$ and $v_n$ are two different nodes in this network, then, $d_{v_m, v_n}$ represents the number of weighted shortest paths between the two nodes, and $d(v_i)_{v_m, v_n}$ particularly measures the number of the weighted shortest paths between the two nodes and crossing node $v_i$.

To test the relationship between geographic distance and international collaboration among nations, we calculate the geographic distance and collaboration strength between country pairs. The geographic distance between nations is defined as distance between capitals of each nation, based on geographic data about world cities in the R package "map". Following the normalization approach used in previous research [6, 24, 25], we apply Salton's cosine measure of international collaboration strength, which takes the publication size of nations into account. It is calculated as the number of collaborative publications divided by the square root of the product of the number of publications of the two collaborating nations (See Appendix Table 2 in S1 Appendix).

## Results

The process of collecting and cleaning the CORD-19 database produced a set of 106,993 publications. The set used for this study is limited to work published in 2020, responding to the search string. We compared the CORD-19 results to Elsevier's Scopus for 2020: the search of Scopus produced 73,000 COVID-19 publications, so fewer than CORD-19. Scopus limits its indexing of publications to specific journals, while CORD-19 encouraged open deposit of materials, which would include venues not indexed by Scopus—this likely accounts for the differences in numbers among databases.

For all research in 2020, Scopus shows a total of 2,584,701 publications, COVID and non-COVID topics (recall that non-COVID topics are not included in CORD-19). Against expectations, growth in life and health sciences output between 2019 and 2020 is shown in most disciplines of life and health sciences fields, both COVID and other topics. The number of publications on novel coronavirus and resulting disease is about 20 times higher in 2020 than coronavirus research published between 2018 and 2019, when work focused on SARS and MERS—earlier disease events that were not as devastating as COVID-19.

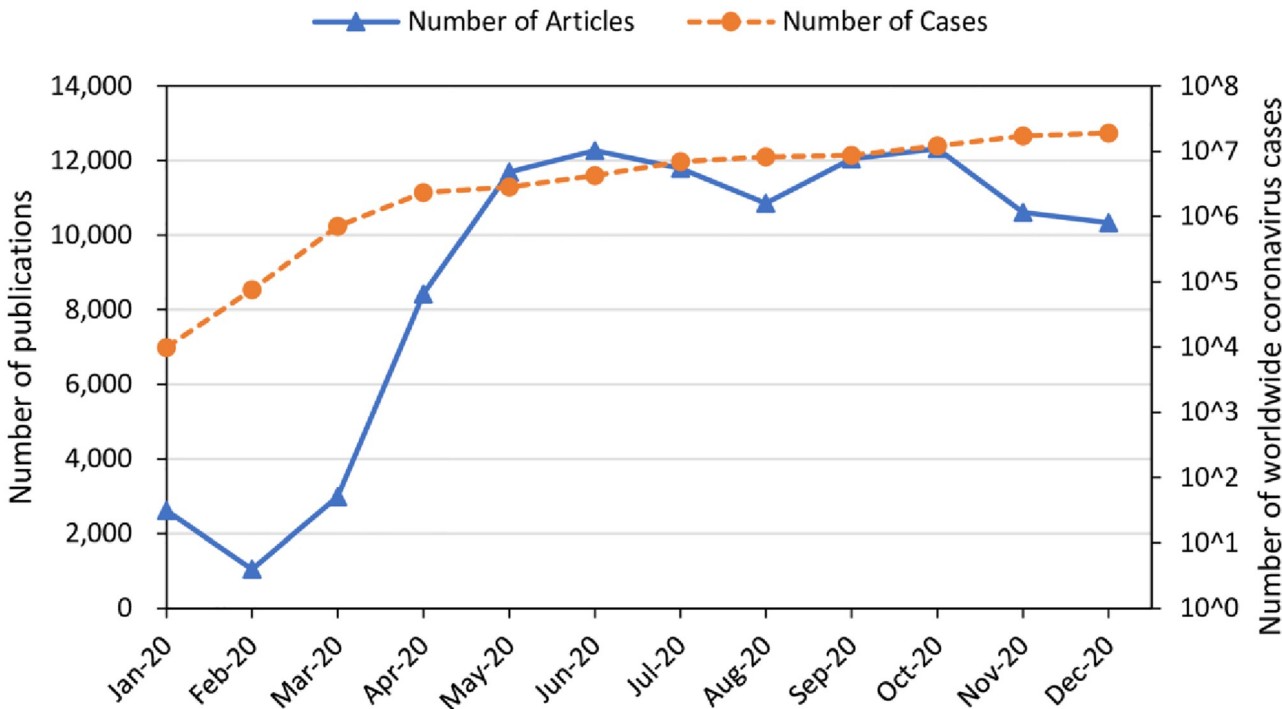

**Fig 1. Number of publications and worldwide COVID-19 cases per month in 2020.** Data on publications and cases are collected from CORD-19 and WHO (https://covid19.who.int/).

Fig 1 shows the number of coronavirus publications by month compared to the number of reported disease case outbreaks worldwide. Publication numbers grew quickly between February and May 2020 at the same time as the number of COVID-19 cases increased at an alarming rate. Since May 2020, the number of publications has remained stable at over 10,000 publications per month, while the rate of growth in COVID-19 cases declined slightly relative to the earliest months. The surge of publications on COVID-19 clearly result from thousands of 'new' researchers from various fields publishing on coronavirus in 2020. In pre-COVID-19 period, coronavirus researchers were drawn mainly from Life Sciences & Biomedical Sciences (e.g., Virology, Infectious Disease) and Natural Sciences (e.g., Multidisciplinary Chemistry and Organic Chemistry). The pandemic calls upon researchers from all research fields, with noticeable increased efforts from Social Sciences (including the authors of this work).

Table 2 shows the top 25 life sciences fields in 2020 from Scopus, with the total number of COVID-19 and non-COVID articles in the same field. Fields that show the highest number of COVID-19 research are medicine, infectious disease, and public health. For all research in life and health sciences, highest growth is seen in surgery, plant science, and psychiatry and mental health. We can assume that the COVID-19 articles were written in 2020, since they are topical —"COVID" was not a keyword in 2019. Moreover, journal editors greatly sped up the processing time for COVID-related review and publication [26, 27]. Conversely, the non-COVID articles may represent work conducted years prior, since it takes time to write, review and publish research results [28]. In fact, peer review in non-COVID related disciplines was delayed in 2020 due to the pandemic [26] so there may be insufficient time to fully assess the impact of the crisis on non-COVID research of publication output. Data in 2021 will be more telling of the impact of the pandemic year on non-COVID research publication patterns.

## Contributions by author location

COVID-19 publishing numbers differ considerably by regions of the world. Fig 2 shows regionally aggregated contributions on COVID-19. Asian countries contributed over one-third of world publications in early 2020, but this percentage share dropped in the later months of 2020 as China reduced its output. Europe showed the opposite trend: Europe's share of world COVID publications increased since April 2020 and the number remained stable through the latter months of 2020. North America's share of publications increased throughout 2020.

As expected, scientifically advanced nations, including China, account for the majority of COVID-19 publications. Among all nations publishing related work, USA, China, and UK produced (together and separately) 50% of the coronavirus articles during 2020, shown in Table 3. As of early 2021, their publications accumulated around 68% of citations made to global publications supporting the expectation of consolidation around expertise and reputation. In the earliest days of the pandemic, three articles from Chinese authors [29–31] contributed key findings that guided much of the ensuing research; each of these articles garnered thousands of citations. Italy, UK, India, and Spain were slower to begin publishing but became more prolific through 2020, and particularly so in the final quarter. Fig 3 shows the rapid growth of monthly publications for selected countries, which also tracks with the trend in

**Table 2. Top 25 fields publishing research on COVID and non-COVID topics.** Data: Elsevier's Scopus.

| Subject | COVID papers 2020 | Non-COVID papers 2019 | Non-COVID papers 2020 | Percentage change non-COVID papers (2019–2020) |
|---|---|---|---|---|
| Biochemistry | 885 | 76492 | 83451 | 9.10 |
| Medicine (all) | 10421 | 71926 | 74548 | 3.65 |
| Surgery | 3043 | 55756 | 67059 | 20.27 |
| Molecular Biology | 1128 | 59901 | 64764 | 8.12 |
| Multidisciplinary | 2070 | 57695 | 61822 | 7.15 |
| Public Health, Environmental and Occupational Health | 5440 | 55134 | 58344 | 5.82 |
| Oncology | 1550 | 48460 | 55061 | 13.62 |
| Ecology, Evolution, Behavior and Systematics | 268 | 50375 | 53869 | 6.94 |
| Biochemistry, Genetics and Molecular Biology (all) | 2214 | 48420 | 52109 | 7.62 |
| Neurology (clinical) | 2139 | 39873 | 44645 | 11.97 |
| Genetics | 647 | 42517 | 43873 | 3.19 |
| Plant Science | 91 | 37264 | 43108 | 15.68 |
| Food Science | 244 | 36888 | 41296 | 11.95 |
| Cardiology and Cardiovascular Medicine | 2493 | 36514 | 41022 | 12.35 |
| Cell Biology | 678 | 36808 | 39631 | 7.67 |
| Agricultural and Biological Sciences (all) | 944 | 37527 | 38366 | 2.24 |
| Psychiatry and Mental Health | 2746 | 33805 | 38290 | 13.27 |
| Cancer Research | 816 | 33733 | 37638 | 11.58 |
| Radiology, Nuclear Medicine and Imaging | 1670 | 33126 | 36210 | 9.31 |
| Pharmacology | 1220 | 34218 | 34576 | 1.05 |
| Infectious Diseases | 5360 | 32048 | 34188 | 6.68 |
| Biotechnology | 492 | 30648 | 33492 | 9.28 |
| Animal Science and Zoology | 130 | 29545 | 33184 | 12.32 |
| Agronomy and Crop Science | 136 | 31115 | 33067 | 6.27 |
| Pediatrics, Perinatology and Child Health | 2043 | 28157 | 32124 | 14.09 |

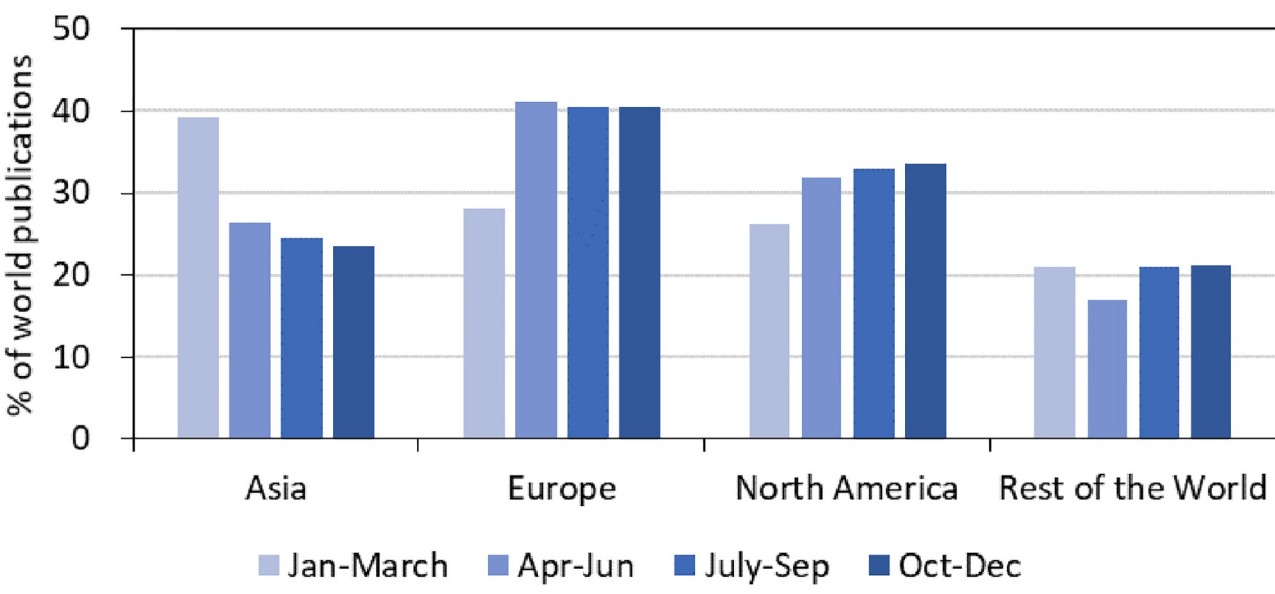

**Fig 2. Author location by region based on 2020 coronavirus publications.**

national COVID-19 cases; this finding is similar to one found in the 2014 West African Ebola epidemic [9].

International collaborative publication rates in coronavirus research took six months to recover to pre-COVID levels. Collaborative research projects take longer to publish results, so the 'recovery' time may simply reflect more communication and production time needed due to the physical distances and time zone differences. As expected, during 2020, international

**Table 3. Productivity of top 10 producers of COVID-19 research in 2020.**

|  |  | Share of total global articles | | Share of total global internationally collaborative articles | |
| --- | --- | --- | --- | --- | --- |
|  |  | **2018–2019** | **2020** | **2018–2019** | **2020** |
|  | Total number of global articles | 5,175 | 92,008 | 1,628 | 20,203 |
| 1 | USA | 1741 (33.6%*) | 26711 (29%) | 856 (52.6%) | 8544 (42.3%) |
| 2 | China | 1240 (24%) | 11591 (12.6%) | 363 (22.3%) | 3259 (16.1%) |
| 3 | UK | 262 (5.1%) | 10744 (11.7%) | 213 (13.1%) | 5164 (25.6%) |
| 4 | Italy | 172 (3.3%) | 8981 (9.8%) | 85 (5.2%) | 2976 (14.7%) |
| 5 | India | 189 (3.7%) | 5950 (6.5%) | 87 (5.3%) | 1684 (8.3%) |
| 6 | Germany | 146 (2.8%) | 4422 (4.8%) | 105 (6.4%) | 1442 (7.1%) |
| 7 | Canada | 308 (6%) | 4116 (4.5%) | 204 (12.5%) | 1990 (9.9%) |
| 8 | France | 228 (4.4%) | 4058 (4.4%) | 134 (8.2%) | 2250 (11.1%) |
| 9 | Spain | 220 (4.3%) | 4115 (4.5%) | 154 (9.5%) | 1629 (8.1%) |
| 10 | Australia | 195 (3.8%) | 3496 (3.8%) | 123 (7.6%) | 2132 (10.6%) |
| 11 | Brazil | 127 (2.5%) | 2736 (3%) | 68 (4.2%) | 919 (4.5%) |

* Percentage share of articles with at least one article from the focal country in total global articles. Top 10 producers in pre-COVID-19 and COVID-19 (11 countries in total) are shown.

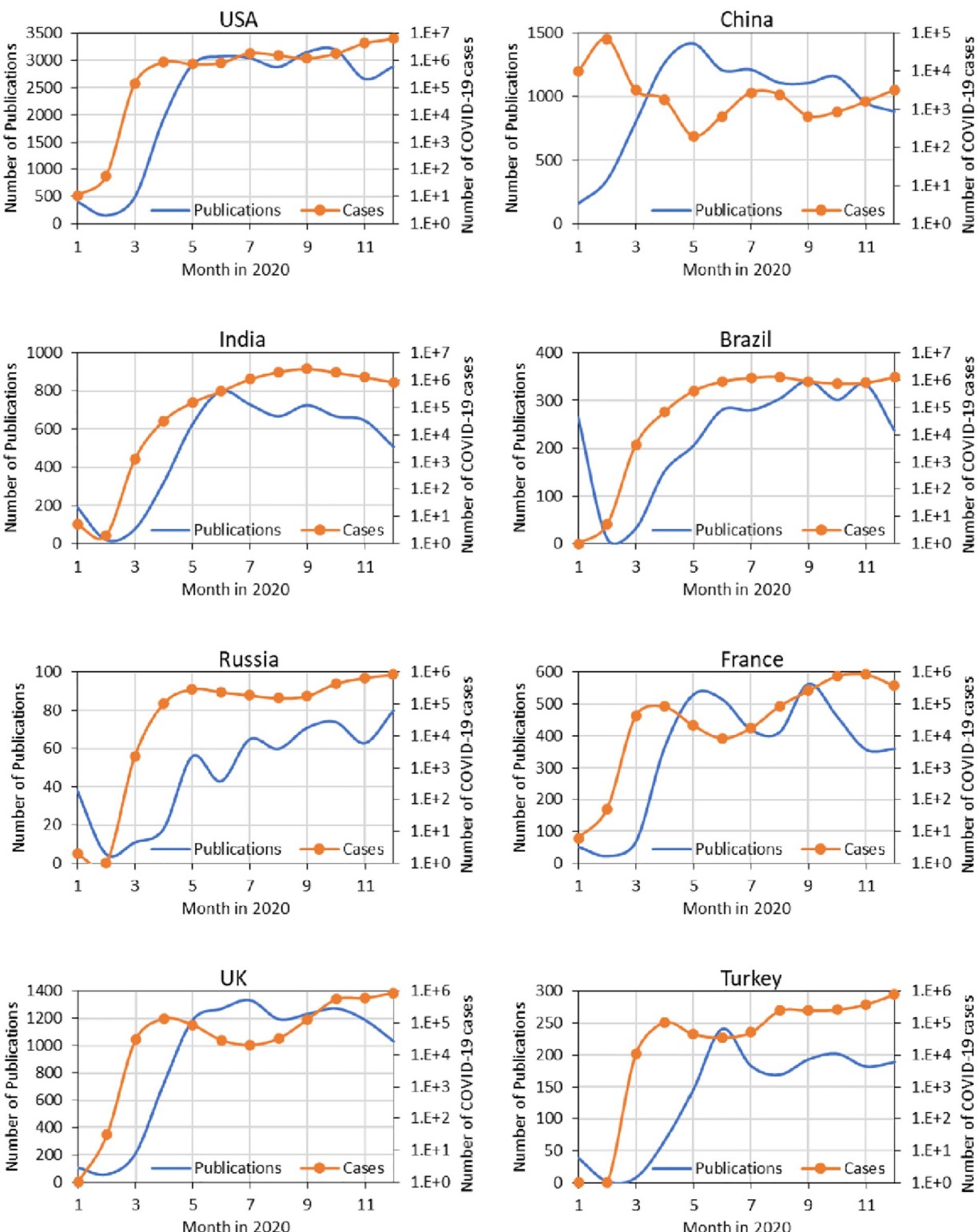

**Fig 3. Number of publications and COVID-19 cases for selected countries by month in 2020.**

collaborative papers showed fewer nations per paper in the early days, but this number increased through the year. This number stabilizes in the fourth quarter to pre-pandemic levels. Supporting other findings [18], about 65% of internationally coauthored papers include only two nations—this is a drop from usual patterns.

In earlier work, we noted that developing nations were largely absent from the publication records in the early COVID-19 period. We explored the participation of developing nations in global coronavirus research over the full year, expecting to see some recovery, but it was weak. Pre-COVID-19 coronavirus research in 2018–2019 shows that low-income nations [32] accounted for 26% of all nations participating in the research, publishing 4% of global articles. This drops during 2020: In the first two quarters of 2020, low-income nations accounted for 21% of active nations and produced 3.4% of global articles (Low-income countries (LIS) are defined by the World Bank, https://data.worldbank.org/country/XM. China, India and Brazil are not low-income countries.). That said, throughout 2020 low-income nations increase their contribution to the coronavirus research, contributing just slightly more in number of publications compared to their participation in pre-COVID-19 period, but much lower than scientifically advanced nations. Against expectations and in contrast to the trend before the pandemic and in the first few months of 2020, we find, by mid-year, Chinese institutions no longer appear in the list of top 10 producing institutions, supporting Liu et al. [18]. For example, the University of Hong Kong and the Chinese Academy of Agricultural Sciences ranked third and fourth in pre-COVID-19 research but dropped down the list in 2020. This drop tracks with the drop in number of COVID-19 cases in China.

Academic institutions worldwide were responsible for the largest share of publications about coronavirus during the pandemic, although private companies participated in research, usually through coauthorship with academic coauthors. We identified a list of 40,287 institutions involved in coronavirus research with the following rules: (1) we retrieved valid institution names with a list of key strings, such as "hospital", "univers*", and "instit*"; and (2) we consolidated variations of the same institutions, such as "MIT" and "Massachusetts Institute of Technology", and "University of Sydney" and "Sydney University". Fig 4 shows the institutions making top contributions to COVID-19 cooperation. As expected, the figure shows that highly reputed institutions—Harvard University (Massachusetts, USA), Huazhong University of Science and Technology (Wuhan, China), and the University of California System—produced the largest absolute numbers of publications on coronavirus in 2020. From the CORD-19 data, we find there are 2,232 articles (2.43%) involving authors from private corporations, which is average for corporate participation in Web of Science [33]. Nevertheless, this percentage is a drop from private sector participation in pre-COVID-19 dataset, where we found that 3.4% of articles involved the private sector, so it was higher than average and dropped to average in 2020.

Table 4 shows the most frequently acknowledged funding agencies in coronavirus research indexed in Web of Science in 2020. Funders from the USA, China, and UK (or Europe) are the most commonly acknowledged, which is consistent with the publication outputs as shown in Table 3. The US National Institutes of Health (NIH), the largest scientific organization dedicated to health and medical research, tops the list and is acknowledged in 15.8% of funded articles. The dominant funder in China, National Natural Science Foundation of China (NSFC), ranks second and contributes to 10.4% of publications, despite decreased publication shares in later periods. European funding agencies, including European Commission, UK Research & Innovation (UKRI), and Medical Research Council UK (MRC), also play a vital role as COVID-19 cases and number of publications increase in Europe.

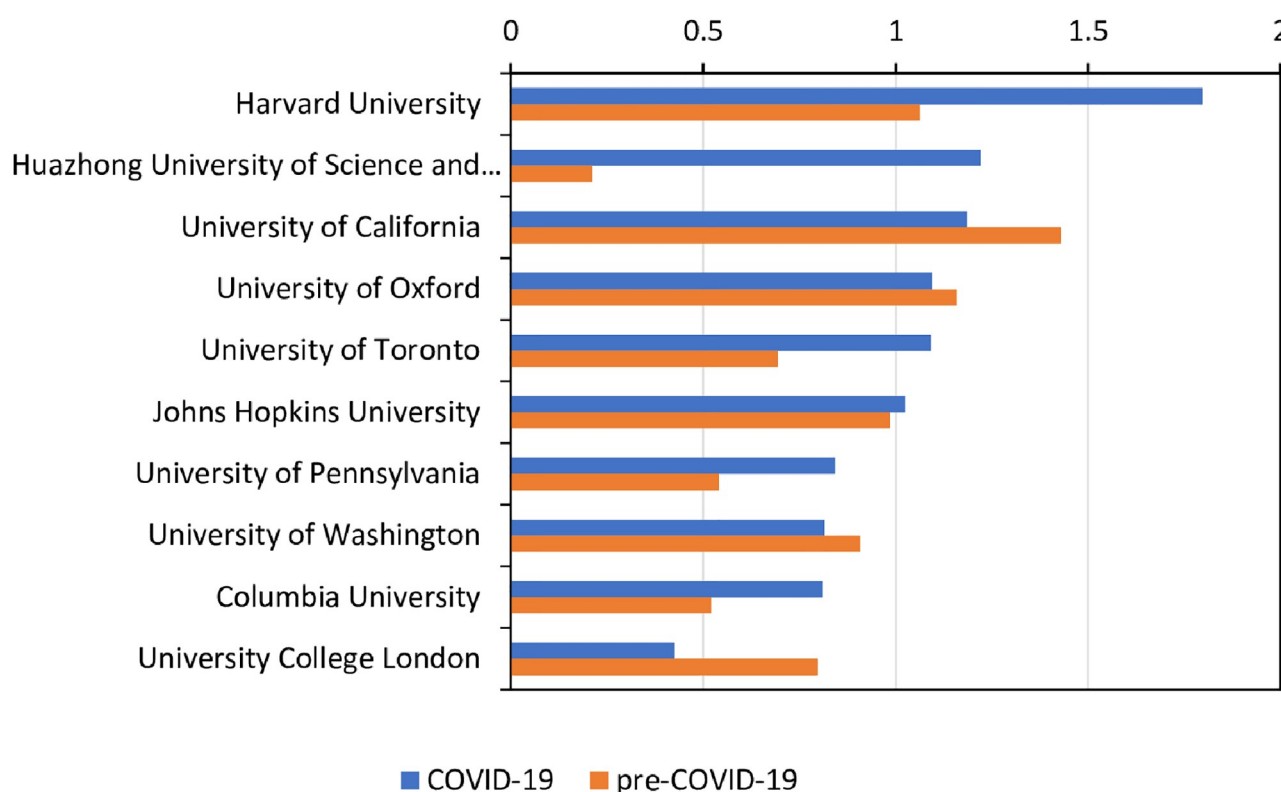

**Fig 4. Top institutions in COVID-19, 2020.** The percentage shares of publications in global articles of top 10 prolific institutions in COVID-19 (2020) and pre-COVID-19 (2018–2019) are shown.

## Co-occurrence network on coronavirus research

Terms retrieved from titles and abstracts of research articles provide useable clues to understand topic focus (Data for network analysis is available at Wagner, Caroline (2021): Figshare_-Network files.rar. figshare. Dataset. https://doi.org/10.6084/m9.figshare.16652752.v1). The co-occurrence between terms and entities (e.g., funding agencies), and among terms, reveals their semantic connections in research, and may answer questions such as which agencies support which topics. The connection is measured by how many times two terms appear in proximity in the entire dataset, within and across articles. We identified 4,865 terms from a raw set of 1.2 million terms retrieved from titles and abstracts of the 106,993 articles published in 2020 via

**Table 4. Major funders of COVID-19 research in 2020.**

|  | Jan–Mar | Apr–June | July–Sep | Oct–Dec | Overall 2020 |
|---|---|---|---|---|---|
| Number of funded articles | 1,224 | 7,090 | 9,996 | 10,487 | 28,797 |
| National Institutes of Health (NIH)—USA | 174 (14.2%) | 1247 (17.6%) | 1,585 (15.9%) | 1,537 (14.7%) | 4,543 (15.8%) |
| National Natural Science Foundation of China (NSFC) | 264 (21.6%) | 916 (12.9%) | 947 (9.5%) | 865 (8.2%) | 2,992 (10.4%) |
| European Commission | 66 (5.4%) | 451 (6.4%) | 659 (6.6%) | 706 (6.7%) | 1,882 (6.5%) |
| UK Research & Innovation (UKRI) | 35 (2.9%) | 222 (3.1%) | 306 (3.1%) | 358 (3.4%) | 921 (3.2%) |
| Medical Research Council UK (MRC) | 26 (2.1%) | 172 (2.4%) | 209 (2.1%) | 238 (2.3%) | 645 (2.2%) |

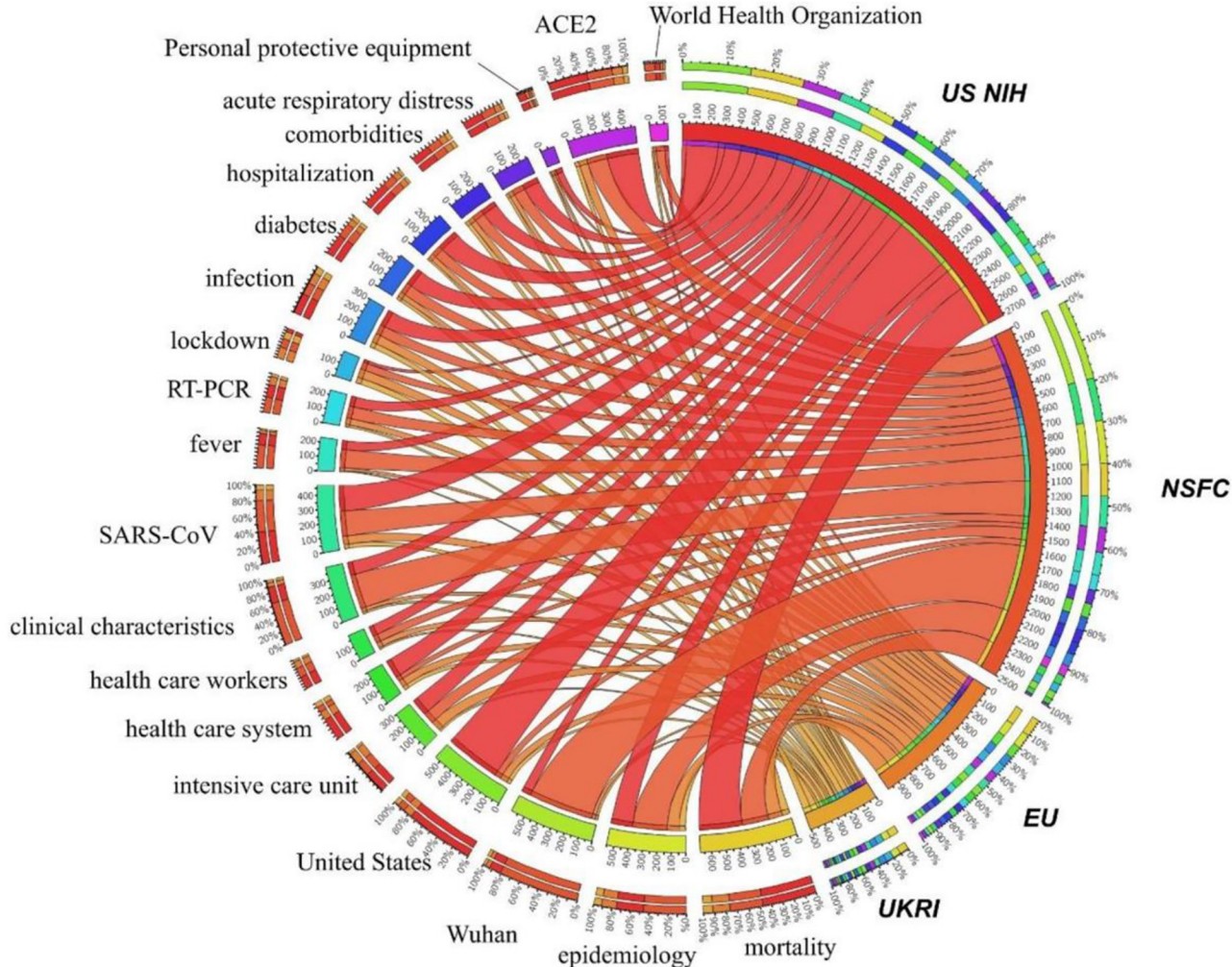

**Fig 5. COVID-related topics funded by major public funding agencies, 2020.**

natural language processing techniques and a term clumping process [34], with the aid of VantagePoint (VantagePoint is a text mining tool for analyzing bibliometric data. See the link: https://www.thevantagepoint.com/). Specifically, the term clumping process facilitated a set of thesauri to remove meaningless terms (e.g., conjunctions, pronouns, and prepositions) and common terms in academic articles (e.g., "method" and "conclusion"), and it then consolidated terms with the same stem (e.g., terms in singular and plural forms).

Fig 5 analyzes the co-occurrence between the top 20 high-frequency terms and the major funding sources, visualized by Circos [35]. The Chinese agency, National Science Foundation of China (NSFC), was much more likely to fund research related to "Wuhan" while the United States' National Institutes of Health (NIH) is much more likely to fund research with the term "United States." Small differences can be observed for "clinical characteristics" (proportionately more from NSFC), and "hospitalization" (proportionately more from NIH) but these two terms are quite similar, so differences may be due to semantics only. Aside from these two differences, it appears that each of the agencies fund similar term portfolios differentiated only in proportion to their contribution, so more focused on basic research, which was our expectation.

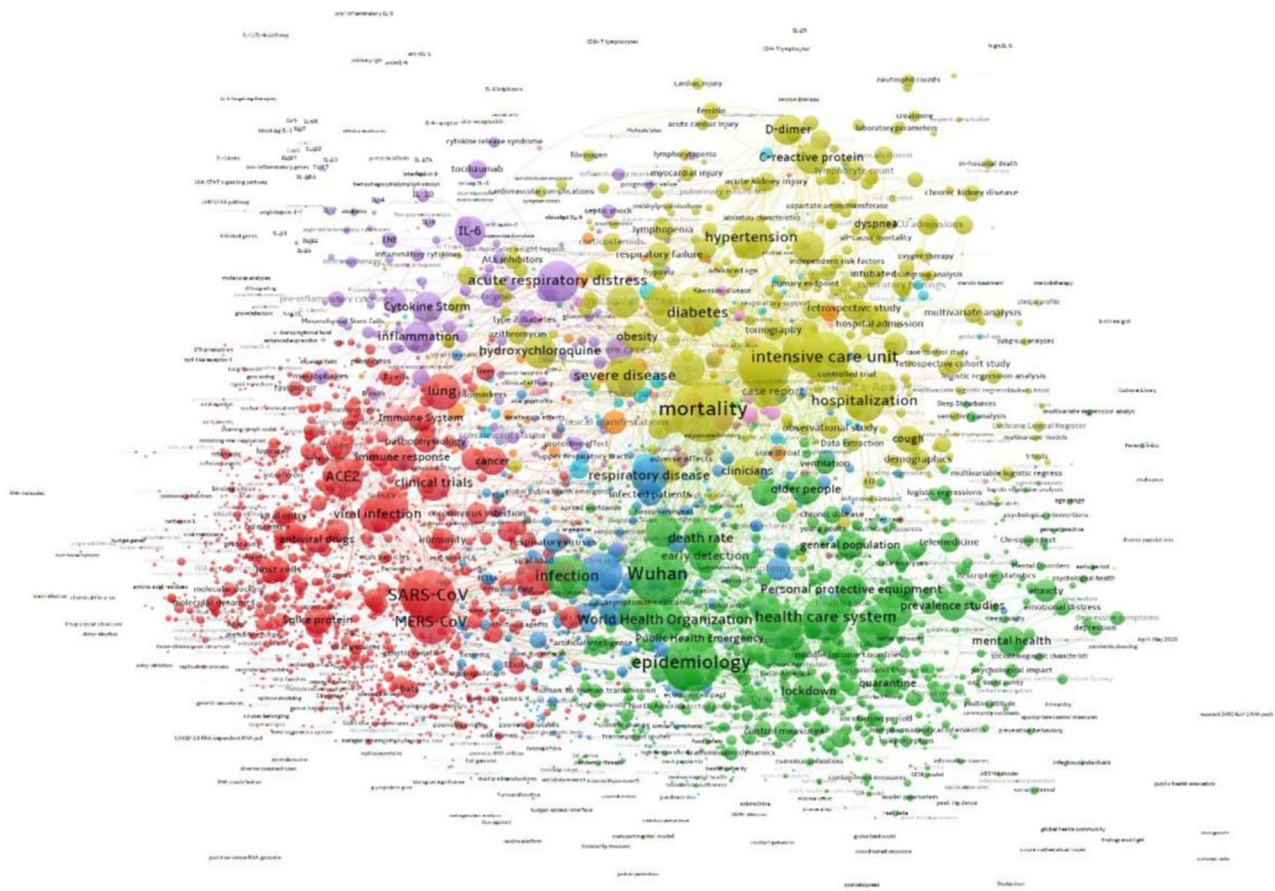

**Fig 6. Topic network of international collaborative research on COVID, 2020.** Interactive version accessible at https://app.vosviewer.com/?json=
https://drive.google.com/uc?id=1MVWE1bsTGi6jJeeU7BNCcjKTd2yjTiv0.

To assess whether international collaborations had different topic focus from domestic collaborations—which we expected—we analyzed co-terms at both levels. Fig 6 shows the co-term network for the internationally collaborative research, and Fig 7 shows the co-term network for domestic-only collaborations. In both networks, one sees the topics that are shown in Fig 5 as focus areas for government funders. With data extracted using Vantagepoint's Natural Language Processing function ported into VOSViewer, Fig 6 shows international collaboration dominated by three clusters: 1) research on the virus (red, bottom left), 2) on patient care (purple, top right), and 3) on public health (green, bottom left).

Fig 7 shows domestic topics, highlighting greater emphasis on patient care and disease characteristics (gold, top center) than seen in Fig 6. Moreover, a fourth cluster emerges (blue, center) with details about outbreaks, effects, viral loads, and other aspects of health are seen that are not as prevalent at the international level. A table in the S1 Appendix provides more details about the topics. As expected, the domestic publications focus on public health and patient care more than is seen at the international level, where basic science dominates the topics.

## Collaboration rate

Table 5 compares collaboration rates in coronavirus publications before the COVID-19 pandemic and in four quarters of 2020. As expected, and in comparison to the number of co-

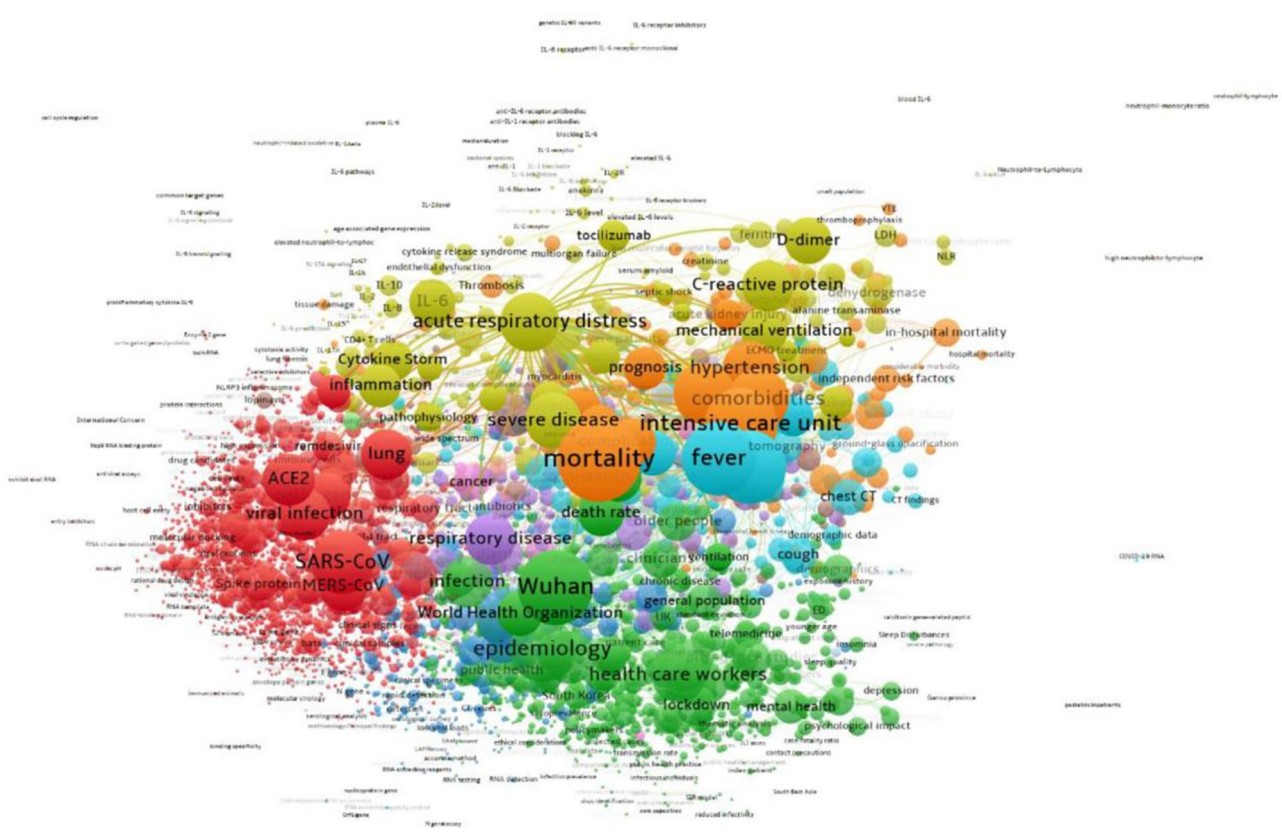

**Fig 7. Topic clusters for domestic research on COVID-19, 2020.** Interactive version accessible at https://app.vosviewer.com/?json=https://drive. google.com/uc?id=1RraBpIYbLY5_DfOMC0YJ7IZoq3_sRiKa.

authors in pre-COVID-19 coronavirus research, publications in 2020 show fewer authors, fewer nations per paper, and less frequent international collaboration overall. Team size—represented by the number of authors on a publication—shrank shortly after the outbreak of COVID-19, a finding we highlighted in Fry et al. [1], but by the end of the year, the number had recovered and risen to just above levels seen in pre-COVID-19 coronavirus research. As expected, the average number of nations per international publication remained at lower levels than pre-COVID-19 levels for USA articles; there was no significant difference in numbers of international partners for Chinese articles in pre-COVID-19 and COVID-19 periods.

We explored the relationship between team structure as represented by coauthors on papers and the number of citations to COVID-19 publications for publications produced in 2020 (looking at citation records up until March 2021). Table 5 shows the regression results that explore the relationship between citations to publications and international collaborative team structure, for publications with authors from USA, China, and the UK respectively. As expected, a positive correlation is shown between numbers of citations and international collaboration. Further, also meeting expectations, there is a correlation between number of authors and citations, which may reflect an audience affect due to a larger reader network. Also as expected, international teams attracted more citations than domestic-only teams, again, with a possible audience affect. These findings held for all nations except the USA, where international articles are not cited more than domestic-only research when holding constant the number of authors (Table 6, column 6).

**Table 5. Collaboration rate in coronavirus research.**

| | Pre-COVID-19 [a] | COVID-19, 2020 | | | | |
|---|---|---|---|---|---|---|
| | | Total | Jan-March | Apr-Jun | July-Sep | Oct-Dec |
| **In Total Articles** | | | | | | |
| Number of Authors | 7.62 (5.69) | 6.91* (20.87) | 6.25* (8.23) | 6.08* (8.30) | 6.91* (16.16) | 7.78 (31.95) |
| Number of Nations | 1.49 (1.00) | 1.36* (0.84) | 1.30* (0.75) | 1.34* (0.84) | 1.37* (0.87) | 1.37* (0.84) |
| International Participants | 0.32 (0.46) | 0.22* (0.41) | 0.19* (0.39) | 0.21* (0.41) | 0.23* (0.42) | 0.23* (0.42) |
| **In USA articles** | | | | | | |
| Number of Authors | 8.34 (7.20) | 7.06* (15.20) | 5.44* (5.17) | 6.15* (7.15) | 7.07* (18.27) | 8.05 (17.54) |
| Number of Nations | 1.81 (1.23) | 1.54* (1.02) | 1.63* (1.03) | 1.55* (1.05) | 1.54* (1.01) | 1.53* (1.00) |
| International Participants | 0.49 (0.50) | 0.32* (0.47) | 0.39* (0.49) | 0.32* (0.46) | 0.32* (0.47) | 0.31* (0.46) |
| **In European articles** | | | | | | |
| Number of Authors | 8.56 (0.18) | 8.15 (0.16) | 7.07* (0.32) | 6.75* (0.10) | 8.32* (0.21) | 9.42 (0.45) |
| Number of Nations | 2.03 (0.04) | 1.63* (0.01) | 1.63* (0.03) | 1.61* (0.01) | 1.64* (0.01) | 1.64* (0.01) |
| International Participants | 0.56 (0.01) | 0.35* (0.00) | 0.35* (0.013) | 0.33* (0.00) | 0.35* (0.00) | 0.35* (0.00) |
| **In Chinese articles** | | | | | | |
| Number of Authors | 8.66 (4.70) | 8.17 (12.8) | 7.46* (6.05) | 7.86* (6.17) | 8.66 (21.59) | 8.31 (6.23) |
| Number of Nations | 1.43 (0.95) | 1.46 (0.92) | 1.33* (0.75) | 1.40 (0.92) | 1.54* (0.97) | 1.52* (0.93) |
| International Participants | 0.29 (0.46) | 0.28 (0.45) | 0.21* (0.41) | 0.25* (0.44) | 0.31 (0.42) | 0.32 (0.47) |

Standard deviation in parentheses.

[a] Based on data used in Cai et al. [15] article.

* denotes statistical significance at p values of 0.05 in a difference of means test comparing pre-COVID-19 and COVID-19 outcomes. Comparisons are between pre-COVID-19 outcomes and outcomes in different COVID-19 quarters.

Data source: CORD-19.

## Network analysis

Figs 8 and 9 compare internationally collaborative networks in the pre-COVID-19 (2018–2019) and COVID-19 (2020) periods. Recall that these numbers represent about 22% of all COVID research in 2020. Fig 8 shows pre-COVID coronavirus research with two large clusters: one, a European cluster, and two, a global cluster brokered by the USA. The US

**Table 6. Negative binomial regression analysis of the relationship between team structure and citation impact of coronavirus publications in 2020.**

| | World total | | | USA | | | China | | | UK | | |
|---|---|---|---|---|---|---|---|---|---|---|---|---|
| | (1) | (2) | (3) | (4) | (5) | (6) | (7) | (8) | (9) | (10) | (11) | (12) |
| Number of citations | | | | | | | | | | | | |
| International collaboration | 0.409*** | | 0.195*** | 0.406*** | | 0.077 | 0.260*** | | 0.195*** | 0.587*** | | 0.258*** |
| | (0.052) | | (0.030) | (0.077) | | (0.058) | (0.074) | | (0.065) | (0.085) | | (0.067) |
| Ln(number of authors) | | 0.668*** | 0.653*** | | 0.682*** | 0.671*** | | 0.787*** | 0.780*** | | 0.779*** | 0.740*** |
| | | (0.029) | (0.028) | | (0.038) | (0.040) | | (0.057) | (0.057) | | (0.046) | (0.044) |
| Journal | Fixed | | | Fixed | | | Fixed | | | Fixed | | |
| Published month | Fixed | | | Fixed | | | Fixed | | | Fixed | | |
| N | 90412 | 90412 | 90412 | 26445 | 26445 | 26445 | 11445 | 11445 | 11445 | 10499 | 10499 | 10499 |

Cluster robust standard errors (cluster on journal) in parentheses.

* $p < 0.1$,

** $p < 0.05$,

*** $p < 0.01$.

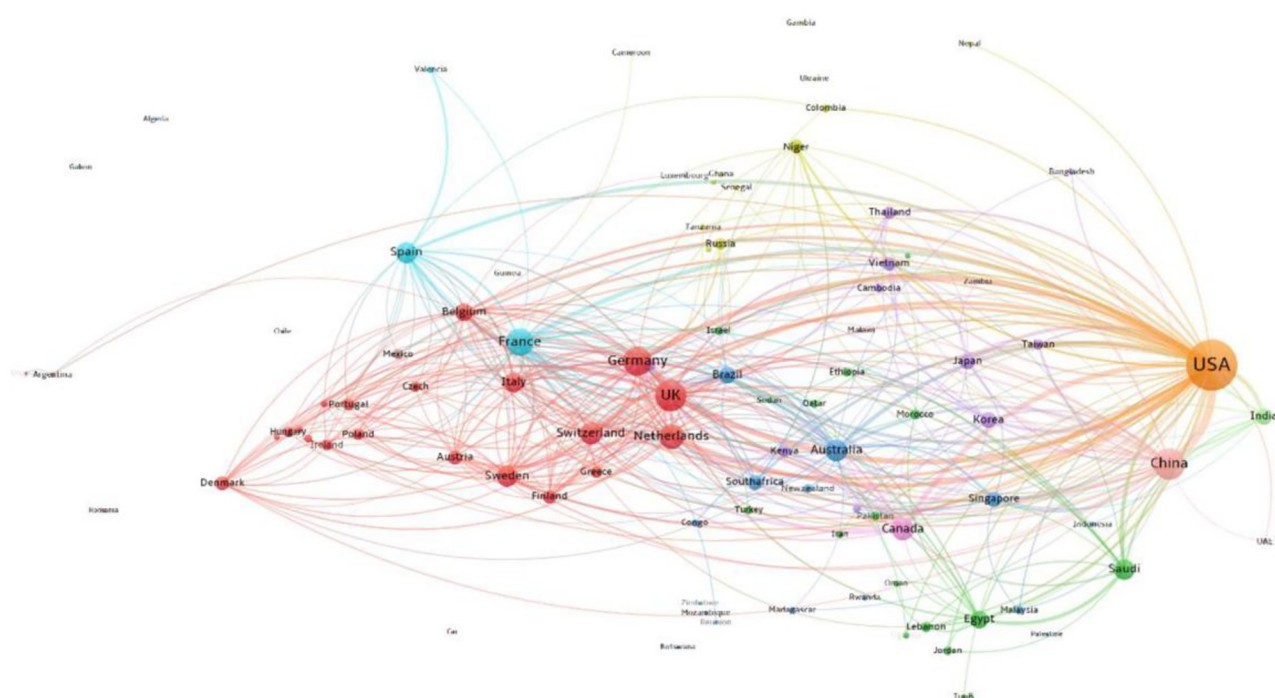

**Fig 8. Pre-COVID international collaboration network in coronavirus research (2018–2019).** Interactive version accessible at https://app.vosviewer.com/?json=https://drive.google.com/uc?id=1_1ASbt-FheQE6_VQNc5eFhy578jot9co.

collaborated closely with China, the UK, France, the Netherlands, and Germany. The dominance of the USA is partly accounted for by the volume of research when compared to the output per nation; the USA leads in publications, citations, and connectivity in coronavirus research prior to the pandemic and remained the leader during the pandemic.

Fig 9 shows the COVID-19 collaborative network, where, as expected based upon research by Rotolo & Frickel [11], we observe more clusters and more brokering hubs than pre-COVID-19. The number of clusters has grown, with four clusters revealing a broader set of countries acting as centralized nodes or hubs, with the UK, Italy, and Germany increasing their bridging role from positions shown in the pre-COVID-19 network. The European research clusters form into two large groups, one with Italy as the brokering hub and one with the UK in a central brokering hub. Italy intensively links to France, the USA, and Switzerland. African nations join the network through the UK connection. Australia is central to a cluster that includes Spain, Brazil, and many smaller nations from South America. As expected, geographic distance between country pairs is negatively related to the collaboration strength of the two countries in all cases (see Appendix Table 2 in S1 Appendix). However, during COVID-19 period, the negative impact of geographic distance on collaboration strength was weakened as demonstrated by positive and significant coefficient of interaction of COVID and geographic distance. As expected, physical distance was less of a barrier to collaboration than in other scientific research. The result reveals that the pandemic weakened the role of geographic distance in international collaboration (OLS regression analysis of the relationship between geographic distance and collaboration strength).

Table 7 shows the network metrics for the above networks and for four quarters of 2020. (Visuals of the four quarterly networks are shown in the S1 Appendix) Consistent with the growing number of internationally collaborative articles throughout 2020, the network

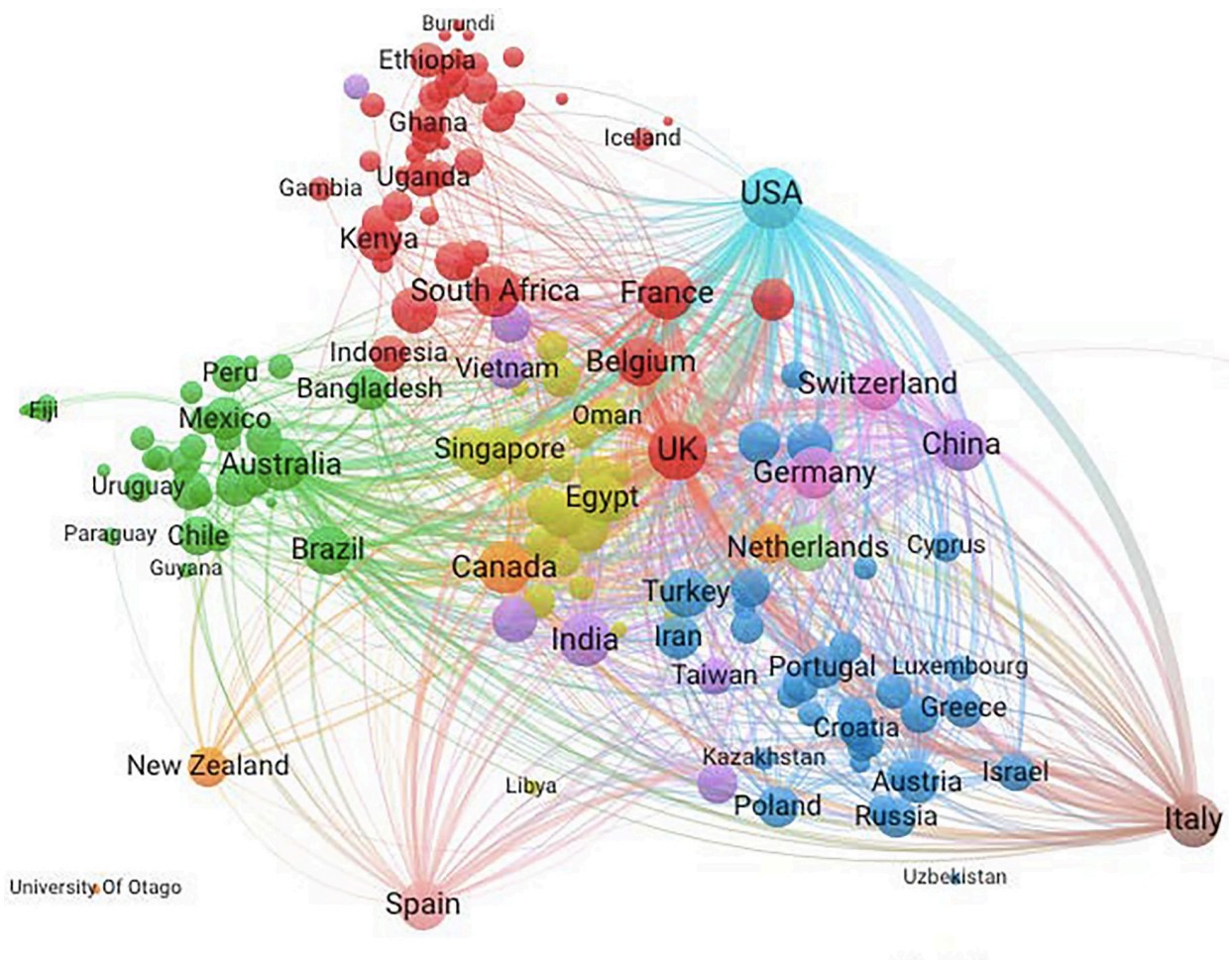

**Fig 9. COVID-19 international collaboration network in coronavirus research, 2020.** Interactive version accessible at https://app.vosviewer.com/?json=https://drive.google.com/uc?id=1TLdcW-lNUQ1k0kDlpZYOUQL57E8MKNqM.

statistics reveal expanded connections from the first period (January to March) to the third period, cumulative (July to September) in 2020, but then stabilizing.

The network statistics suggest that COVID-19 research involved many more participants than those who worked on coronavirus in the years prior to the pandemic, as we find by examining the number of unique author names. From the pre-COVID-19 coronavirus network to the COVID-19 network, we see that number of nodes increases from 103 nations before

**Table 7. Network metrics in coronavirus research.**

|  | Pre-COVID-19 | COVID-19, 2020 | | | | |
|---|---|---|---|---|---|---|
|  |  | Jan–Mar | Apr–June | July–Sep | Oct–Dec | Overall 2020 |
| Number of International collaborative articles | 1628 | 918 | 5,704 | 6,978 | 6,631 | 20,231 |
| Number of nodes | 103 | 109 | 151 | 164 | 169 | 173 |
| Number of links | 1147 | 732 | 2,054 | 2,524 | 2,604 | 3,796 |
| Average degree | 22.3 | 13.4 | 27.2 | 30.8 | 30.8 | 43.9 |
| Network density | 0.218 | 0.124 | 0.181 | 0.189 | 0.183 | 0.255 |

**Table 8. Network metrics for USA, UK, and China, pre-COVID and COVID.**

| | Pre-COVID-19, 2018–2019 | | | COVID-19, 2020 | | |
|---|---|---|---|---|---|---|
| | USA | UK | China | USA | UK | China |
| Number of International collaborative articles | 856 | 213 | 363 | 8,553 | 5,171 | 3,265 |
| Degree | 92 | 65 | 55 | 158 | 146 | 117 |
| Weighted betweenness centrality | 0.932 | 0.099 | 0.088 | 0.868 | 0.270 | 0.022 |

COVID to 173 during COVID. More importantly, the number of links among nations more than triples, suggesting that many more connections were made at the international level than existed prior to the pandemic. Average degree doubles, supporting the observations of many new links. These links likely were forged remotely in a process that Liu et al. [18] call "parachuting collaborations" that post-date the pandemic. These types of collaborations may have emerged through friend-of-friend connections, since people could not meet face-to-face due to travel restrictions during 2020. Betweenness centrality drops over the year indicating a shift in influence of the initial, dominant hubs to include more participants from more nations over the pandemic year.

Table 8 shows the network metrics for top producers (USA, UK, and China) in the global network. As shown by the average degree of the three countries, the USA was a hub in the network in both periods, more so early in the pandemic, supporting our expectation of consolidation around expertise and reputation, but betweenness centrality drops as researchers from more countries joined into the research. The UK played a much more active role in the COVID network compared to China in the second half of the year. Despite being a hub in the pre-COVID network and in the first months of the pandemic [1], China played a less prominent role in the network in 2020.

## Limitations of this research

This research project had a number of limitations of data, time, analysis, and scope. Data limitations include constraints of what is measurable in published work (publications, networks, and citations). We decided to use CORD-19 data because it was the most expansive dataset for COVID-19 research, but we may have picked up lower quality work as a result; it is an open dataset with attendant problems. We extracted desired features, but there may be gaps and errors. In order to get a count of open-access publications, we used Dimensions data, but the total number of COVID-19 publications in Dimensions were lower than CORD-19, so open-access publications are likely under-counted. We present the percentages of COVID-19 research that is open access; these calculations are broadly representative, but they cannot be further verified. Moreover, we are unable to show extent of R&D occurring in private research laboratories if it is not published. We hoped to inform policymakers about COVID-19 research trends in a timely manner, which meant we worked with data available at the time (in Spring 2021) rather than waiting until data has been expanded, cleaned or validated. Elsevier and Clarivate databases were also examined; these databases are more carefully curated for quality. We tapped them for comparisons to CORD-19, and especially for non-COVID research. We had planned to test whether the pandemic had an impact on non-COVID research but we were unable to show this outcome: During 2020, peer reviews were delayed [36], researchers were not able to access labs or other resources, and scholarship was interrupted, but these obstacles are not yet evident in the data. Disruptions to 2020 research activities likely will not be seen in publication data until 2022 and after. We also exclude preprint

publications from this analysis, which could present a limitation, given the important role of preprints during the pandemic.

Further, a limitation of this analysis is the reliance on nations as 'super-nodes' in a network that consists of individuals with associated cultural contexts that are not captured in network data. This reliance on nations is partly justified in that nations represent the underlying political and social systems that support scientific activities by offering funding, infrastructure, training, and dissemination of results. We acknowledge that the reliance on nations as a unit of measurement is a limitation, imposed on the analyst based upon the ways in which data are collected. Future research will need to ensure cleaner data to validate comparisons presented here. None of these datasets can truly represent the scope of activities that contributed to what is known about COVID-19, and mechanisms to assess knowledge flows are quite limited and time consuming to collect.

## Discussion

A review of one year of research publications about the novel coronavirus that emerged in Wuhan China in late 2019 shows the research community reacting rapidly and robustly to the challenge. The rapidity of response to COVID-19 suggests flexibility in the research system: thousands of researchers from many fields began working on the crisis. Research was disseminated initially in a flood of preprints; rapidly peer-reviewed publications were placed on open data platforms or shared openly on subscription-based platforms. COVID-related, peer-reviewed publications rose sharply in number in early 2020, and these publications were much more likely to be shared on open-access platforms or formats to enable rapid knowledge diffusion than is the norm in scholarly publishing [17]. The earlist COVID-19 research efforts were conducted by China, the USA, and the UK, and these three nations constitute close to 50% of all COVID publications on the subject in 2020. European nations started off slowly in research publications, but these nations continued to grow their output throughout 2020, as China's output dropped.

As expected, international collaborations accounted for a smaller percentage of publications than is generally seen in 'normal' times, where internationally co-authored articles often account for more than one-quarter of articles [put new footnote here that was added on page 4 and delete this note] We expected the drop-off in international publications because a lack of mobility meant that people were unable to meet and discuss shared insights, or to devise, carry out or compare research results. Remote collaborations involve higher transaction costs and could be expected to slow progress. This possibly explains why international teams were smaller: to cut down on the time needed for communication. Further, the lower rate of international collaboration may be due to topics related to patient care and public health specific to particular regions or nations rendering them less suited to international collaboration [3]. We also noted that distance was less of a barrier to collaboration than is shown in other studies in times before the COVID-19 crisis.

Interestingly, the number of papers per nation tracks closely with the outbreak of COVID-19 cases in that nation. We expected to find numbers of publications to be more closely correlated to research funding. This may still be the case, but the data is too variable and incomparable to elicit a correlation between funding and output. We surmise that researchers were motived by a desire to be helpful to those suffering with the disease. It may also be the case that local COVID cases provided observational opportunities for researchers, and thus publication opportunities, as well, which produced data that resulted in more national publications.

The low rate of participation by lower-income nations was somewhat unexpected. Lower-income nations had a very low rate of participation in the early days of COVID, and only

slowly joined the global publication counts. This may be due to a number of factors, including the need to publish locally to address the crisis, the inability of researchers to access laboratories or data during lock-down periods, the lack of access to one's office, or the inability of national ministries to provide emergency research funds. People may not have had access to the Internet at home. The lower showing for developing nations is a concern, since these nations need the scientific knowledge to battle viral events just as much or more than advanced nations. This finding clearly requires more research and perhaps policy action.

We expected that the combined rush to work on COVID and the pandemic lock-downs would reduce non-COVID research activities. This may still be the case (reported in a survey by Myer et al. [2]), but it could not be detected in publication numbers at this writing. Publications in life and health sciences in non-COVID topics increased in number over 2019. As researchers become more comfortable with remote work, they may have persisted in publishing earlier results, however, this does not comport with the findings of Myer et al. [2]. As the pipeline catches up these activities, it will be worth revisiting the impact on productivity again at the end of 2021.

The stratification and consolidation comport with a model of global science as a reputation-based system creating a social hierarchy: the global network reverts to scientifically advanced nations and elite institutions in a crisis. We expected that the number of papers would align with reputation and resources. The role of reputation is confirmed by the consolidation of actors to fewer, elite institutions in scientifically advanced nations cooperating together more so than prior to the pandemic. The role played by access to resources is unclear —we observe that national output is closely tied to number of COVID-19, which could be due to a desire of scientists to help the effort. This requires more inquiry.

The influence of geopolitical factors also appears to play some role in research output, partnership and productivity. Arguably, Chinese publications initiated most of the COVID-19 research into the nature of the SARS-CoV-2 virus itself [29–31]. Nevertheless, in April 2020, the Chinese government changed requirements for review of articles related to the origins of COVID-19, requiring a more central review for work about the source of the novel coronavirus, which may have reduced the willingness of Chinese authors to cooperate internationally, although this requires more inquiry. Negative political comments made in the United States against China regarding the source of the virus may also have dampened international collaboration, although we did not test for this possibility.

Time and attention may have played a role in the drop in the share of internationally co-authored papers. Transaction costs of distance communications may have hampered some international connections. The drop-off in number of developing nations participating at the start of the pandemic may have contributed to the drop in international collaboration numbers, as well, by lowering the number of potential collaborators.

A remaining question arises around the mechanisms by which people, who had not already worked together before the pandemic, became connected to one another in a year when most people were physically isolated from each other. These connections are what Liu et al. [18] term "parachuting collaborations." One would expect that people connect face-to-face: Research shows that the vast majority of collaborative projects start face-to-face or side-by-side. When that cannot take place, it is unclear whether people look for physically proximate partners, choose to work alone, become connected to new people through friend-of-a-friend, through social media, or perhaps just a 'cold-call' outreach to someone they do not know. It is clear that many of the connections made around COVID-19 may not have existed prior to the pandemic, so further research is needed to understand how people connected with each other under crisis conditions.

## Supporting information

**S1 Appendix. Network visuals.**
(DOCX)

## Acknowledgments

Thanks go to Clayton E. Tillman and Thomas Collins for help with data collection and formatting.

## Author Contributions

**Conceptualization:** Caroline S. Wagner, Yi Zhang, Caroline V. Fry.

**Data curation:** Caroline S. Wagner, Xiaojing Cai, Caroline V. Fry.

**Formal analysis:** Caroline S. Wagner, Xiaojing Cai, Yi Zhang, Caroline V. Fry.

**Investigation:** Xiaojing Cai.

**Methodology:** Xiaojing Cai, Caroline V. Fry.

**Project administration:** Caroline S. Wagner.

**Software:** Yi Zhang.

**Validation:** Yi Zhang.

**Visualization:** Yi Zhang.

**Writing – original draft:** Caroline S. Wagner, Caroline V. Fry.

**Writing – review & editing:** Xiaojing Cai, Caroline V. Fry.

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
