## [Decision Letter · Decision Letter 0]

2 Sep 2021

PONE-D-21-22845

One Year of COVID-19 Research at the International Level in CORD-19 Data

PLOS ONE

Dear Dr. Wagner,

Thank you for submitting your manuscript to PLOS ONE. After careful consideration, we feel that it has merit but does not fully meet PLOS ONE’s publication criteria as it currently stands. Therefore, we invite you to submit a revised version of the manuscript that addresses the points raised during the review process.

We look forward to receiving your revised manuscript.

Kind regards,

Alberto Baccini, Ph.D.

Academic Editor

PLOS ONE

Journal Requirements:

 [NO]. 

3. We note you have included a table to which you do not refer in the text of your manuscript. Please ensure that you refer to Table 1 in your text; if accepted, production will need this reference to link the reader to the Table.

Reviewers' comments:

Reviewer's Responses to Questions

**Comments to the Author**

1. Is the manuscript technically sound, and do the data support the conclusions?

Reviewer #1: Yes

Reviewer #2: Partly

2. Has the statistical analysis been performed appropriately and rigorously? 

Reviewer #1: Yes

Reviewer #2: No

3. Have the authors made all data underlying the findings in their manuscript fully available?

Reviewer #1: Yes

Reviewer #2: No

4. Is the manuscript presented in an intelligible fashion and written in standard English?

Reviewer #1: Yes

Reviewer #2: Yes

5. Review Comments to the Author

Reviewer #1: It is a timely update about the COVID research in 2020 with details on international collaboration. Authors have reported the changes and compared with pre-covid, covid 2019, 2020 on several dimensions: ranking of country, institutions, co-word analysis, funding, publication and cases. It is important to have such thorough update about COVID research in 2020.

The author claimed that coronavirus research is nearly doubled from 2008 to 2018. It will be great to see whether other topics in PubMed have higher growth rate than coronavirus. Or provide one example of the growth rate of other domains, or what is the average growth rate of PubMed so that we can have a better idea of the growth rate of coronavirus research when we have a baseline to compare with. Another thing is that coronavirus before 2019 are all regional, there are no significant funding invested in this domain, even include SARS in China. Therefore, there might not be many international efforts on coronavirus research before 2019. In page 5 second paragraph, the last sentence, please provide the evidences for the statement here that US authors were most likely to collaborate with researchers from China, UK, Germany by either citing an article or providing data points.

Since most of the epidemics are regional, is the research collaboration also regional driven that advanced nations collaborate with scientists from the regions where epidemics happened. Also based on the spreading areas of epidemics, the international collaboration patterns might change that more regions of spread areas will join the international collaboration. Like the example the author proposed here Ebola, the regions who joined the international collaboration can be connected with the areas of Ebola spreads. Also when Eloba dies down, the international collaboration regions will shrink, and also reduced. These ideas might be interesting for authors to explore here in COVID. The relationship between the spread time and regions might lead to the increase or decrease of collaborations of advanced countries with such regions.

Second reasons could be political reasons in different countries, which might lead to the significant changes in the landscape of international collaboration in COVID. Figure 1 is very telling, would be great to include also the number of non COVID publications as the third line to contrast the trend. Also, how do you define a non-COVID article? In Table 2, is that non-COVID paper is the whole PumMed publication in that year minus the number of COVID articles? What is Multidsiciplinary in Subject from Table 2 means here? You said that your data for Table 2 is Elsevier data, you mean scopus data?

In figure 2, it will be great to know the latin amarica, especially countries like Brazil. Also the changes of nations are related to the spread regions of diseases as diseases first started from asian, then to europe, then to US, then to south america. It will be great to see the distribution of collaborative COVID papers as national and international, and their changes here in Figure 2. Or the increase of cases in a country and its international collaborated papers whether it incresses along with the cases or have different trends, or the international collaboration starts earlier than the first case of a country?

In page 19, the authors talked about the low-income countries only contribute slightly more COVID publications compared to their pre-COVID period. But in Table 3, china, india and brazil are ranked top 11 countries according to the productivity. Are china, india, and brazil considered as the low-income countries? What is your definition of low-income countries?

For Figure 4 institution ranking, can you please explain the normalization of institution names, which can be tedious. Do you use any tools or what rules you follow to clean and merge institution names.

In figure 5, how do you identify 4,865 terms, do you choose them based on the ranked frequency? do you exclude the frequent terms, do you merge terms if they are similar, such as one is in a singular form, and one is a plural form? Also words from figure 5 come from different categories, some are cities, some are bio entities, some are generic terms. Do you think it is possible to show different figures based on the same types of terms. For example, you can do the same figure but only with top keywords of bio entities, other figures could be keywords about countries, the other could be keywords about social events, such as lockdowns.

Reviewer #2: This paper presents a bibliometric analysis of the scientific literature on COVID-19. The analysis is mostly sound and clear, although I have some concerns about the network analysis. Below I provide more detailed comments.

“A section on data and methodology presents experiments designed to answer … A results section describes outcomes of the experiments”: Instead of ‘experiments’, my suggestion is to use ‘analyses’. The term ‘experiments’ is confusing, because the analyses presented in the paper do not represent experiments in the way this term is commonly understood in most fields of science.

“Global production of research articles in biology and biomedical sciences, of which coronavirus research is a subset, nearly doubled from 165,000 in 2008 and to 306,000 in 2018. The largest percent increases came from lower, lower-middle, and upper-middle income nations. [6]”: Ref. [6] is from 2004, so this reference cannot be the source of statistics for 2008 and 2018. Please add the correct reference.

“In 2018, at about 20,000, the United States was the most prolific producer of life and health sciences publications”: The source of this statement is unclear. Since this statement is located in the literature review section, I assume it is based on earlier literature, but no reference is provided. The number of 20,000 publications seems unrealistically low to me. I think the number should be much higher.

“Most of the emergency funds went to national institutions, although the European Union (EU) and the US National Institutes of Health (NIH) fund both national and foreign applicants … Most nations did not publish early pandemic research”: This text appears twice in the literature review section, in almost identical ways.

“COVID-19 publications were much more likely than other works to be published as open access in 2020 [15], also known as ‘gold OA’ papers.”: Publications that were published as open access are not necessarily called gold OA. They are called gold OA only if there are openly accessible in a journal (rather than in a repository or on a preprint server).

“were more likely than other work to be published in subscription-based journals such as The Lancet, Science, New England Journal of Medicine or Nature but these works were placed into open Web portals for rapid access, called ‘green OA’.”: This statement is inaccurate. Many subscription-based journals have made their COVID-19 articles openly accessible in the journal. This is called gold OA, not green OA. There are also subscription-based journals (e.g., Elsevier journals) that have made their COVID-19 articles openly accessible in PubMed Central, while the articles have not been made openly accessible on the journal website. This is indeed called green OA.

“The initial dataset was cleaned to remove the following artifacts: conference papers, preprints, collections of abstracts, symposia results, articles pre-dating 2020, and meeting notes.”: Why did you exclude preprints? They are often seen as a major innovation in scholarly communication resulting from the pandemic.

“four quarters according to ‘Published Date’”: How is the published date defined? Is this the date on which an article was published online, or is this the official date of publication of the journal issue in which an article is included?

“complete dataset of scientific articles”: A data set based on PubMed, Scopus, and Web of Science should not be called complete, since these are selective databases that do not provide a full coverage of the scholarly literature.

“For articles indexed in Clarivate’s Web of Science (WoS), we retrieved funding information using PubMed ID”: This is confusing. Web of Science includes funding information for all publications that acknowledge funding, regardless of whether these publications are indexed in PubMed or not. I therefore don’t understand why you use PubMed IDs.

“VOSviewer developed by Waltman et al. [19]”: This is not the right reference. Please refer to https://doi.org/10.1007/s11192-009-0146-3.

“Salton’s measure is applied”: To properly correct for the size of countries, you need to use a different measure, sometimes called the association strength (https://doi.org/10.1002/asi.21075) or the probabilistic affinity (or activity) index (https://doi.org/10.1023/A:1005632319799). See also https://doi.org/10.1177%2F016224399201700106 and https://doi.org/10.1007/BF02016282.

“Comparing CORD-19 to Elsevier’s database for 2020”: It is not clear what you mean by ‘Elsevier’s database’. In the data and methodology section, you mention a database obtained by combining data from PubMed, Scopus, and Web of Science, but you do not mention a database that includes only Elsevier data.

Please clarify whether the results presented in the results section are based on a full (whole) counting approach or a fractional counting approach for dealing with co-authored publications. Although I have the impression that a full counting approach is used, this is not entirely clear. For instance, I wonder whether the statistics reported in Table 3 are based on full or fractional counting.

I struggle to understand Figure 5. The figure is hard to interpret and doesn’t seem to have a clear added value. My suggestion is to remove the figure from the paper.

Figures 6 and 7: I wonder whether VOSviewer’s text mining features were used to extract terms from titles and abstracts or whether you used your own term extraction process as discussed in the first paragraph of the section ‘Co-occurrence network on coronavirus research’. Please clarify this.

Table 5: There is no need to report the statistics with three decimals. One or two decimals is sufficient. It is not clear to me what the numbers between parentheses represent. Also, I am not sure what is meant by “Based on data used in SCIM autumn update article”.

Figures 8 and 9: The figures are hard to read. Consider increasing the size of the labels in VOSviewer. In addition, you may consider making interactive visualizations available online. This can easily be done using the ‘Share’ button in the most recent version of VOSviewer.

“The growth of clusters may reflect broadening of subjects in the topic-focus” and “The increased clustering may reflect the emergence of new topics during the pandemic year”: This is quite speculative. The increase in the number of clusters is small, from three clusters to four clusters, and the clustering results produced by VOSviewer are likely to be quite sensitive to the value of the so-called resolution parameter (available on the ‘Analysis’ tab in VOSviewer). Moreover, the increase in the number of clusters may also be due to an increase in the number of countries included in the collaboration network.

Table 7: It is not clear what we learn from this table. The table needs to be explained and interpreted in a proper way.

Tables 8 and 9: I am skeptical about many of the network metrics reported in these tables (e.g., network density, average path length, betweenness centrality, average clustering coefficient). These metrics are highly complex, and interpreting them in a sensible way is challenging. Giving a proper interpretation to these metrics is especially difficult for weighted networks, like the collaboration networks studied by the authors, since many metrics were originally developed for unweighted networks, not for weighted ones. Also, comparing the values of the network metrics obtained for the different time periods is problematic because the number of nodes (i.e., countries) in the networks is not stable. Instead of reporting a broad set of network metrics, my suggestion is to start from the questions you want to answer about the collaboration networks and to then identity the relevant network metrics needed to answer these questions. Only these metrics need to be presented. The others don’t need to be included.

“we suspect that the non-COVID research activities were not fully represented in 2020”: This is unclear. What do you mean by this?

“we observe that national output is more closely tied to number of cases than it is to financial resources”: The paper doesn’t seem to provide sufficient empirical evidence to support this conclusion.

6. PLOS authors have the option to publish the peer review history of their article (what does this mean?). If published, this will include your full peer review and any attached files.

Reviewer #1: No

Reviewer #2: **Yes: **Ludo Waltman

---

## [Author Response · Author response to Decision Letter 0]

14 Oct 2021

September 20, 2021

Dear Editors,

Thank you for the peer review comments you provided to us for the draft manuscript, “One Year of COVID-19 Research at the International Level in COVD-19 Data”. We have edited and amended the manuscript to respond to peer reviewers’ comments. The responses are described in this letter and they are incorporated into the final draft manuscript we are submitting with this letter.

One of the reviewers noted that the data for the study needs to be available online. A figshare site has been established to provide the cleaned dataset of all collaborations to any researcher who wishes to check our work or build upon it with the same data. The doi is: https://doi.org/10.6084/m9.figshare.16620274.v1 and it is noted in the paper. 

Comment:

The author claimed that coronavirus research is nearly doubled from 2008 to 2018. It will be great to see whether other topics in PubMed have higher growth rate than coronavirus. Or provide one example of the growth rate of other domains, or what is the average growth rate of PubMed so that we can have a better idea of the growth rate of coronavirus research when we have a baseline to compare with. Another thing is that coronavirus before 2019 are all regional, there are no significant funding invested in this domain, even include SARS in China. Therefore, there might not be many international efforts on coronavirus research before 2019. In page 5 second paragraph, the last sentence, please provide the evidences for the statement here that US authors were most likely to collaborate with researchers from China, UK, Germany by either citing an article or providing data points.

Response:

This finding was in the first paper that the coauthors wrote (Fry et al., 2020) in which we discussed coronavirus research growth. Doubling matches the field of biology as a whole, which also doubled in that time. Moreover, we cannot support a finding that coronavirus research was regionally focused: it has had an international component since the first days after SARS. The paragraph with the sentence cited by the reviewer was removed—these data were from the National Science Foundation. This was not our work, and we could not answer the questions about it, so we removed it from the paper. 

Comment:

Since most of the epidemics are regional, is the research collaboration also regional driven that advanced nations collaborate with scientists from the regions where epidemics happened. Also based on the spreading areas of epidemics, the international collaboration patterns might change that more regions of spread areas will join the international collaboration. Like the example the author proposed here Ebola, the regions who joined the international collaboration can be connected with the areas of Ebola spreads. Also when Ebola dies down, the international collaboration regions will shrink, and also reduced. These ideas might be interesting for authors to explore here in COVID. The relationship between the spread time and regions might lead to the increase or decrease of collaborations of advanced countries with such regions.

Response from authors:

We thank the reviewer for this suggestion for future research.

Comment:

Second reasons could be political reasons in different countries, which might lead to the significant changes in the landscape of international collaboration in COVID. Figure 1 is very telling, would be great to include also the number of non COVID publications as the third line to contrast the trend. Also, how do you define a non-COVID article? In Table 2, is that non-COVID paper is the whole PubMed publication in that year minus the number of COVID articles? What is Multidisciplinary in Subject from Table 2 means here? You said that your data for Table 2 is Elsevier data, you mean Scopus data?

Response from authors:

We thank the reviewer for this suggestion. The non-COVID publications in 2020 cannot be directly compared to the COVID publications. Our assessment is that the COVID work was published rapidly in response to the crisis. Non-COVID publications were actually delayed for a number of reasons. The two data sets are not directly comparable. 

We identified the data from Elsevier as coming from Scopus, although we obtained the data directly from Elsevier, not through the Scopus interface. 

Comment:

In figure 2, it will be great to know the latin amarica, especially countries like Brazil. Also the changes of nations are related to the spread regions of diseases as diseases first started from asian, then to europe, then to US, then to south america. It will be great to see the distribution of collaborative COVID papers as national and international, and their changes here in Figure 2. Or the increase of cases in a country and its international collaborated papers whether it incresses along with the cases or have different trends, or the international collaboration starts earlier than the first case of a country?

Response from authors:

We thank the reviewer for this comment and question. Brazil appears as a coauthoring country in about 250 records, so it would not show up in Figure 2. The reviewer is welcome to view the original data at https://doi.org/10.6084/m9.figshare.16620274.v1 on figshare to search for Brazil’s collaborations. 

Comment:

In page 19, the authors talked about the low-income countries only contribute slightly more COVID publications compared to their pre-COVID period. But in Table 3, china, india and brazil are ranked top 11 countries according to the productivity. Are china, india, and brazil considered as the low-income countries? What is your definition of low-income countries?

Response from authors:

This section is now on page 17 of the draft, we have used the World Bank list of countries by income. We added an endnote about this fact. China, India and Brazil are not low-income countries.

Comment:

For Figure 4 institution ranking, can you please explain the normalization of institution names, which can be tedious. Do you use any tools or what rules you follow to clean and merge institution names.

Response from authors:

Using machine learning, we cleaned and merged institution names based on the following rules: (1) we retrieved raw affiliation names with a list of key strings, such as "hospital“, "univers*", "college", "inc", and "institut*"; (2) we consolidated variations of the same institutions, such as "university of sydney" and "Sydney U", "MIT" and "Massachusetts Institute of Technology".

Comment:

In figure 5, how do you identify 4,865 terms, do you choose them based on the ranked frequency? do you exclude the frequent terms, do you merge terms if they are similar, such as one is in a singular form, and one is a plural form? Also words from figure 5 come from different categories, some are cities, some are bio entities, some are generic terms. Do you think it is possible to show different figures based on the same types of terms. For example, you can do the same figure but only with top keywords of bio entities, other figures could be keywords about countries, the other could be keywords about social events, such as lockdowns.

Response from authors:

The identification of the 4,865 terms was based on the term clumping process (Ref[27]), which is a semi-automatic process for removing noise terms and consolidating synonyms. Regarding your specific questions: (1)Yes, you are right. We chose the top terms in Figure 5 based on frequency. (2) Yes, we have a set of thesauri for term removal and consolidation. Briefly, we removed common and general terms, as well as those common terms in academic articles (e.g., "introduction" and "method"), and we also merged terms with the same stem, as you mentioned the same terms in the singular form and the plural forms. The key idea of Figure 5 is to see the diverse emphases of major public funding agencies in COVID topics, and thus, despite that we can categorize these terms into different sectors and create a series of such figures, considering the key focus of this paper is 'international collaboration' and we prefer to use such a figure to stand at a relatively macro level to compare the different interests of these funding agencies. However, conducting in-depth topic analyses to understand those detailed topics is among our future directions.

Reviewer #2: This paper presents a bibliometric analysis of the scientific literature on COVID-19. The analysis is mostly sound and clear, although I have some concerns about the network analysis. Below I provide more detailed comments.

Comment:

“A section on data and methodology presents experiments designed to answer … A results section describes outcomes of the experiments”: Instead of ‘experiments’, my suggestion is to use ‘analyses’. The term ‘experiments’ is confusing, because the analyses presented in the paper do not represent experiments in the way this term is commonly understood in most fields of science.

Response from authors:

Thank you for this comment, we have made the suggested change.

Comment:

“Global production of research articles in biology and biomedical sciences, of which coronavirus research is a subset, nearly doubled from 165,000 in 2008 and to 306,000 in 2018. The largest percent increases came from lower, lower-middle, and upper-middle income nations. [6]”: Ref. [6] is from 2004, so this reference cannot be the source of statistics for 2008 and 2018. Please add the correct reference.

Response from authors:

We removed this paragraph since these data were from an external source, not our data, and it was not needed for the argument.

Comment:

“In 2018, at about 20,000, the United States was the most prolific producer of life and health sciences publications”: The source of this statement is unclear. Since this statement is located in the literature review section, I assume it is based on earlier literature, but no reference is provided. The number of 20,000 publications seems unrealistically low to me. I think the number should be much higher.

Response:

We removed this paragraph since these data were from an external source, not our data, and it was not needed for the argument.

Comment:

“Most of the emergency funds went to national institutions, although the European Union (EU) and the US National Institutes of Health (NIH) fund both national and foreign applicants … Most nations did not publish early pandemic research”: This text appears twice in the literature review section, in almost identical ways.

Response from authors:

Thank you for catching this error. We have corrected it in the revised manuscript.

Comment:

“COVID-19 publications were much more likely than other works to be published as open access in 2020 [15], also known as ‘gold OA’ papers.”: Publications that were published as open access are not necessarily called gold OA. They are called gold OA only if there are openly accessible in a journal (rather than in a repository or on a preprint server).

Response from authors:

We removed the references to gold and green OA since it is confusing and not needed for the argument.

Comment:

“were more likely than other work to be published in subscription-based journals such as The Lancet, Science, New England Journal of Medicine or Nature but these works were placed into open Web portals for rapid access, called ‘green OA’.”: This statement is inaccurate. Many subscription-based journals have made their COVID-19 articles openly accessible in the journal. This is called gold OA, not green OA. There are also subscription-based journals (e.g., Elsevier journals) that have made their COVID-19 articles openly accessible in PubMed Central, while the articles have not been made openly accessible on the journal website. This is indeed called green OA.

Response from authors:

We removed the references to gold and green OA since it is confusing and not needed for the argument. We also noted that subscription-based journals offered COVID papers as ‘open’ on their websites.

Comment:

“The initial dataset was cleaned to remove the following artifacts: conference papers, preprints, collections of abstracts, symposia results, articles pre-dating 2020, and meeting notes.”: Why did you exclude preprints? They are often seen as a major innovation in scholarly communication resulting from the pandemic.

Response from authors:

We thank the reviewer for this question. We added a sentence into the paper to explain why we removed preprints from this treatment.

New text:

“The initial dataset was cleaned to remove the following artifacts: conference papers, preprints, collections of abstracts, symposia results, articles pre-dating 2020, and meeting notes. Preprints were excluded in this report to avoid double-counting in cases where a work is subsequently peer-reviewed and published….” 

Comment:

“four quarters according to ‘Published Date’”: How is the published date defined? Is this the date on which an article was published online, or is this the official date of publication of the journal issue in which an article is included?

Response from authors:

We add a note to explain the ‘Published Date” which is the electronic publication date where available or else the print publication date. P. 9 of the revised manuscript.

Comment:

“complete dataset of scientific articles”: A data set based on PubMed, Scopus, and Web of Science should not be called complete, since these are selective databases that do not provide a full coverage of the scholarly literature.

Response from authors:

Thank you for catching this mistake. We have corrected the statement in the methodology section.

Comment:

“For articles indexed in Clarivate’s Web of Science (WoS), we retrieved funding information using PubMed ID”: This is confusing. Web of Science includes funding information for all publications that acknowledge funding, regardless of whether these publications are indexed in PubMed or not. I therefore don’t understand why you use PubMed IDs.

Response from authors:

We used WoS to gather additional data on funding for a subset of our sample that is indexed in WoS. Articles indexed in WoS but not included in our sample are not considered. Therefore, the PubMed IDs are used to match data on funding acknowledgements (if any). This is explained in the revised manuscript.

Comment:

“VOSviewer developed by Waltman et al. [19]”: This is not the right reference. Please refer to https://doi.org/10.1007/s11192-009-0146-3.

Response from authors:

Thank you for correcting this reference. The corrected reference is reflected in the manuscript, reference number 19.

Comment:

“Salton’s measure is applied”: To properly correct for the size of countries, you need to use a different measure, sometimes called the association strength (https://doi.org/10.1002/asi.21075) or the probabilistic affinity (or activity) index (https://doi.org/10.1023/A:1005632319799). See also https://doi.org/10.1177%2F016224399201700106 and https://doi.org/10.1007/BF02016282.

Response from authors:

Thank you for this comment and suggestion. While we appreciate the redirection, we have added citations to the paper on earlier use and validation of the use of Salton’s measure in international analysis. We hope to explore the suggestion made by the reviewer in future research.

Comment:

“Comparing CORD-19 to Elsevier’s database for 2020”: It is not clear what you mean by ‘Elsevier’s database’. In the data and methodology section, you mention a database obtained by combining data from PubMed, Scopus, and Web of Science, but you do not mention a database that includes only Elsevier data.

Response from authors:

We have clarified the multiple data sources. CORD-19 does not have non-COVID publications, and we wanted to compare them, so we used data from Elsevier.

Comment:

Please clarify whether the results presented in the results section are based on a full (whole) counting approach or a fractional counting approach for dealing with co-authored publications. Although I have the impression that a full counting approach is used, this is not entirely clear. For instance, I wonder whether the statistics reported in Table 3 are based on full or fractional counting.

Response from authors:

We did not use VOSviewer's text mining features to extract terms from raw text. Instead, we used VantagePoint's Natural Language Processing function and an integrated term clumping function, which is a semi-automatic process for removing noise terms and consolidating synonyms via thesauri and macros. Briefly, we remove terms such as conjunctions, prepositions, and common terms in academic articles (e.g., "method") and consolidate terms based on their stems (e.g., the same term but in the singular form and the plural form). This is noted in an endnote in the methodology section.

Comment:

I struggle to understand Figure 5. The figure is hard to interpret and doesn’t seem to have a clear added value. My suggestion is to remove the figure from the paper.

Response from authors:

Thank you for your comment on this figure. With all due respect, we request to retain the figure. It shows which agencies of government funded what topics. This information is not available anywhere else in the article and we believe it will be useful to a policy-oriented reader.

Comment:

Figures 6 and 7: I wonder whether VOSviewer’s text mining features were used to extract terms from titles and abstracts or whether you used your own term extraction process as discussed in the first paragraph of the section ‘Co-occurrence network on coronavirus research’. Please clarify this.

Response from authors:

We did not use VOSviewer's text mining features to extract terms from raw text. Instead, we used VantagePoint's Natural Language Processing function and an integrated term clumping function, which is a semi-automatic process for removing noise terms and consolidating synonyms via thesauri and macros.

Comment:

Table 5: There is no need to report the statistics with three decimals. One or two decimals is sufficient. It is not clear to me what the numbers between parentheses represent. Also, I am not sure what is meant by “Based on data used in SCIM autumn update article”.

Response from authors:

Thank you for this suggestion. We left the numbers as 2 decimal places. We added a citation to our earlier work on COVID, published in summer 2021.

Comment:

Figures 8 and 9: The figures are hard to read. Consider increasing the size of the labels in VOSviewer. In addition, you may consider making interactive visualizations available online. This can easily be done using the ‘Share’ button in the most recent version of VOSviewer.

Response from authors:

Thank you for this comment and suggestion. We have added the data to figshare and noted this in the text. We increased the font size of the country names. We are exploring a way to share the visuals for additional inspection. 

Comment:

“The growth of clusters may reflect broadening of subjects in the topic-focus” and “The increased clustering may reflect the emergence of new topics during the pandemic year”: This is quite speculative. The increase in the number of clusters is small, from three clusters to four clusters, and the clustering results produced by VOSviewer are likely to be quite sensitive to the value of the so-called resolution parameter (available on the ‘Analysis’ tab in VOSviewer). Moreover, the increase in the number of clusters may also be due to an increase in the number of countries included in the collaboration network.

Response from authors:

Thank you for this comment. We have reviewed the question of clusters and we agreed that the discussion should be deleted. Clustering can change based upon settings in the data set. At this time, further discussion of clustering does not advance the paper. 

Comment:

Table 7: It is not clear what we learn from this table. The table needs to be explained and interpreted in a proper way.

Response from authors:

Thank you for your request for more explanation. We have added additional text to support the findings of this analysis, and we moved the table to an endnote. New text:

“However, during COVID-19 period, the negative impact of geographic distance on collaboration strength was weakened as demonstrated by positive and significant coefficient of interaction of COVID and geographic distance. As expected, physical distance was less of a barrier to collaboration than in other scientific research. The result reveals that the pandemic weakened the role of geographic distance in international collaboration the endnote.” 

Comment:

Tables 8 and 9: I am skeptical about many of the network metrics reported in these tables (e.g., network density, average path length, betweenness centrality, average clustering coefficient). These metrics are highly complex, and interpreting them in a sensible way is challenging. Giving a proper interpretation to these metrics is especially difficult for weighted networks, like the collaboration networks studied by the authors, since many metrics were originally developed for unweighted networks, not for weighted ones. Also, comparing the values of the network metrics obtained for the different time periods is problematic because the number of nodes (i.e., countries) in the networks is not stable. Instead of reporting a broad set of network metrics, my suggestion is to start from the questions you want to answer about the collaboration networks and to then identity the relevant network metrics needed to answer these questions. Only these metrics need to be presented. The others don’t need to be included.

Response from authors:

We agree with the reviewer that the network metrics shown are unnecessarily complicated and obscure. We simplified the measures by removing more obscure measures and we have focused on density and betweenness measures. We added text in the methodology section about why these measures were chosen. We added clarifying language in the results section. We also removed the discussion of the k-core group as not essential to the line of argument.

Comment:

“we suspect that the non-COVID research activities were not fully represented in 2020”: This is unclear. What do you mean by this?

Response from authors:

We thank the reviewer for a request to clarify this statement. We have added text as follows:

“However, we suspect that the impact on non-COVID research activities are not fully represented in 2020 publication data. Peer reviews were delayed 31, researchers were not able to access labs or other resources, and scholarship was interrupted. As a result, the disruptions to 2020 research activities likely will not be seen in publication data until 2022 and after.” 

Comment:

“we observe that national output is more closely tied to number of cases than it is to financial resources”: The paper doesn’t seem to provide sufficient empirical evidence to support this conclusion.

Response from authors:

We agree with the reviewer on this comment. We have changed the wording as follows:

“Interestingly, the number of papers per nation tracks closely with the outbreak of COVID-19 cases in that nation. We expected to find numbers of publications to be more closely correlated to research funding. This may still be the case, but the data is too variable and incomparable to conduct a test. We surmise that researchers were motived by a desire to be helpful to those suffering with the disease. It may also be the case that local COVID cases provided observational opportunities for researchers and thus publication opportunities, as well, which produced data that resulted in more national publications.”

Again, thank you to the reviewers for these helpful comments which we judge have improved the paper. We endeavored to respond fully to the comments.

With kind regards,

Caroline S. Wagner on behalf of myself and my coauthors

---

## [Decision Letter · Decision Letter 1]

11 Nov 2021

PONE-D-21-22845R1One Year In: COVID-19 Research at the International Level in CORD-19 DataPLOS ONE

Dear Dr. Wagner,

Thank you for submitting your manuscript to PLOS ONE. After careful consideration, we feel that it has merit but does not fully meet PLOS ONE’s publication criteria as it currently stands. Therefore, we invite you to submit a revised version of the manuscript that addresses the points raised by Reviewer #2.

We look forward to receiving your revised manuscript.

Kind regards,

Alberto Baccini, Ph.D.

Academic Editor

PLOS ONE

Journal Requirements:

Reviewers' comments:

Reviewer's Responses to Questions

**Comments to the Author**

1. If the authors have adequately addressed your comments raised in a previous round of review and you feel that this manuscript is now acceptable for publication, you may indicate that here to bypass the “Comments to the Author” section, enter your conflict of interest statement in the “Confidential to Editor” section, and submit your "Accept" recommendation.

Reviewer #1: All comments have been addressed

Reviewer #2: (No Response)

2. Is the manuscript technically sound, and do the data support the conclusions?

Reviewer #1: Yes

Reviewer #2: Partly

3. Has the statistical analysis been performed appropriately and rigorously? 

Reviewer #1: Yes

Reviewer #2: Yes

4. Have the authors made all data underlying the findings in their manuscript fully available?

Reviewer #1: Yes

Reviewer #2: No

5. Is the manuscript presented in an intelligible fashion and written in standard English?

Reviewer #1: Yes

Reviewer #2: Yes

6. Review Comments to the Author

Reviewer #1: (No Response)

Reviewer #2: The authors have made many improvements to their paper. The paper can almost be accepted for publication. I still have a number of remaining comments, mostly of a fairly minor nature.

“The US National Science Foundation (NSF) reports that”: Add a reference.

The discussion of the data and methodology is more clear in the revised paper. I just want to note that it seems to me that the approach taken by the authors, in which Web of Science, Scopus, and Dimensions data were combined, is rather complicated. The use of only one of these three databases instead of all three would probably have been sufficient to carry out all the analyses reported in the paper.

I am unable to access the data sets available at https://figshare.com/account/home#/projects/123277. An email address and password are requested.

Subsection headings don’t seem to have a consistent layout. For instance, ‘Contributions by Author Location’ has a bold formatting, while ‘Co-occurrence network on coronavirus research’ is formatted in italics.

I would like to suggest changing the order of Table 2 and Figure 1. I think it’s more natural to first present Figure 1 and then Table 2.

“We can assume that the COVID-19 articles were written in 2020, since they are topical—“COVID” was not a keyword in 2019”: The second part of this sentence is unclear. What do you mean by saying that ‘COVID’ was not a keyword in 2019? I am also confused because the search terms listed in the Data and Methodology section include the term ‘coronavirus’, and coronavirus research took place not only in 2020 but also in earlier years.

“while the COVID-19 cases also stabilized relative to the earliest months”: This is not correct. The number of cases keeps increasing in Figure 1. Keep in mind that you are using a logarithmic scale. Hence, while the increase may seem small, it is actually a rather large increase.

“However, during COVID-19 period, the negative impact of geographic distance on collaboration strength was weakened as demonstrated by positive and significant coefficient of interaction of COVID and geographic distance”: It is not clear which coefficient of interaction you are referring to.

“Our visual scan indicates that the additional work, over and above the life sciences and biological inquiries, came in the form of public health and patient-care topics specific to the pandemic, which would not have been part of the pre-COVID research.”: Again it is not clear what you are referring to. Which visual scan do you mean?

Tables 7 and 8: If you want to include betweenness in these tables, you need to explain how it was calculated and how it can be interpreted. In Table 8 betweenness is referred to as ‘weighted betweenness’, while in Table 7 it is just called ‘betweenness’, so it is not clear whether the weights of the links between countries were taken into account in the calculation of betweenness or not. A detailed explanation is necessary, both for the calculation and for the interpretation of betweenness. Interpretation of betweenness is challenging especially in weighted networks. Alternatively, you may consider removing betweenness from the tables.

“showing that influence across the network becomes more diffused as researchers from more countries joined into the research”: It is not clear to me what you mean by influence becoming more diffused.

In the limitations section, my suggestion is to mention the exclusion of preprints as a limitation. Preprints have played an important role in the rapid dissemination of COVID-19 research, so their exclusion from your work represents a significant limitation.

“where internationally coauthored articles often account for more than one-third of articles.”: Could you add references to substantiate this statement?

7. PLOS authors have the option to publish the peer review history of their article (what does this mean?). If published, this will include your full peer review and any attached files.

Reviewer #1: No

Reviewer #2: **Yes: **Ludo Waltman

---

## [Author Response · Author response to Decision Letter 1]

2 Dec 2021

November 29, 2021

Editors, PLoS One

Submission: PONE-D-21-22845R1, One Year In: COVID-19 Research at the International Level in CORD-19 Data

Dear Editors,

Thank you for this opportunity to revise the referenced article. We have fully responded to the reviewers’ comments. We appreciate the careful attention given by the reviewers to this work. The referee comments have greatly improved the article. 

Thank you for requesting the reference to the National Science Foundation data. We have included a reference to the National Science Board indicators report.

From referee: The discussion of the data and methodology is more clear in the revised paper. I just want to note that it seems to me that the approach taken by the authors, in which Web of Science, Scopus, and Dimensions data were combined, is rather complicated. The use of only one of these three databases instead of all three would probably have been sufficient to carry out all the analyses reported in the paper.

The reviewer asked why we use three data sets. We apologize if we have not been clear about the data. We have revised the paper to make this clearer. Please note that this team used combined data sets only in the earlier paper, but not in this one. (There, we used preprint and published works, for example.) In the paper currently under review, we considered the use of funding data from the Web of Science because these data were not available from CORD-19. We pulled funding data from the Web of Science and publication data rom CORD-19. We have added text to make these data sets clear.

We sought to correct the figshare link, as requested. We were not able to connect the network data on figshare because the files are too large. The figshare data is the cleaned data from CORD-19.

"Subsection headings don’t seem to have a consistent layout. For instance, ‘Contributions by Author Location’ has a bold formatting, while ‘Co-occurrence network on coronavirus research’ is formatted in italics."

Thank you for this suggestion. We revised the headings. 

"I would like to suggest changing the order of Table 2 and Figure 1. I think it’s more natural to first present Figure 1 and then Table 2."

Thank you for this suggestion - we changed the ordering. 

“We can assume that the COVID-19 articles were written in 2020, since they are topical—“COVID” was not a keyword in 2019”: The second part of this sentence is unclear. What do you mean by saying that ‘COVID’ was not a keyword in 2019? I am also confused because the search terms listed in the Data and Methodology section include the term ‘coronavirus’, and coronavirus research took place not only in 2020 but also in earlier years."

We used the term ‘coronavirus’ for all years; COVID is only used for 2020. This is clarified in the text.

“while the COVID-19 cases also stabilized relative to the earliest months”: This is not correct. The number of cases keeps increasing in Figure 1. Keep in mind that you are using a logarithmic scale. Hence, while the increase may seem small, it is actually a rather large increase."

We changed the phrasing in the text to: "while the rate of growth in COVID-19 cases declined slightly relative to the earliest months"

“However, during COVID-19 period, the negative impact of geographic distance on collaboration strength was weakened as demonstrated by positive and significant coefficient of interaction of COVID and geographic distance”: It is not clear which coefficient of interaction you are referring to."

Thank you for your question. We included an endnote with a link to the table with the coefficients calculated to assess the impact of distance on the likelihood of cooperation.

“Our visual scan indicates that the additional work, over and above the life sciences and biological inquiries, came in the form of public health and patient-care topics specific to the pandemic, which would not have been part of the pre-COVID research.”: Again it is not clear what you are referring to. Which visual scan do you mean?"

We deleted this sentence since it is not critical to the article.

Tables 7 and 8: If you want to include betweenness in these tables, you need to explain how it was calculated and how it can be interpreted. In Table 8 betweenness is referred to as ‘weighted betweenness’, while in Table 7 it is just called ‘betweenness’, so it is not clear whether the weights of the links between countries were taken into account in the calculation of betweenness or not. A detailed explanation is necessary, both for the calculation and for the interpretation of betweenness. Interpretation of betweenness is challenging especially in weighted networks. Alternatively, you may consider removing betweenness from the tables."

We used weighted betweenness centrality in Table 8 to measure the importance of these selected countries in international collaborations. We explained in the text the reason why we chose this indicator and how we calculated; new text is on Page 12 of the revised draft.

“showing that influence across the network becomes more diffused as researchers from more countries joined into the research”: It is not clear to me what you mean by influence becoming more diffused."

We changed the phrasing; "but betweenness centrality drops as researchers from more countries joined into the research"

"In the limitations section, my suggestion is to mention the exclusion of preprints as a limitation. Preprints have played an important role in the rapid dissemination of COVID-19 research, so their exclusion from your work represents a significant limitation."

We added: “We also exclude preprint publications from the analysis which could present a limitation, given the important role of preprints during the pandemic.” 

“where internationally coauthored articles often account for more than one-third of articles.”: Could you add references to substantiate this statement?"

This is changed to one-quarter and a reference to the National Science Board report has been added.

Thank you again for the careful review and thoughtful comments and suggestions. We appreciate it very much.

Caroline S. Wagner on behalf of myself and coauthors

---

## [Editor Report · Decision Letter 2]

7 Dec 2021

One Year In: COVID-19 Research at the International Level in CORD-19 Data

PONE-D-21-22845R2

Dear Dr. Wagner,

We’re pleased to inform you that your manuscript has been judged scientifically suitable for publication and will be formally accepted for publication once it meets all outstanding technical requirements.

Kind regards,

Alberto Baccini, Ph.D.

Academic Editor

PLOS ONE
---

## [Editor Report · Acceptance letter]

14 Jan 2022

PONE-D-21-22845R2 

One-Year In:
COVID-19 Research at the International Level in CORD-19 Data 

Dear Dr. Wagner:

I'm pleased to inform you that your manuscript has been deemed suitable for publication in PLOS ONE. Congratulations! Your manuscript is now with our production department. 

Kind regards, 

on behalf of

Prof. Alberto Baccini 

Academic Editor

PLOS ONE